# Two classes of amine/glutamate multi-transmitter neurons innervate Drosophila internal male reproductive organs

**Marta Chaverra[1], John Paul Toney[1], Lizetta D Dardenne-Ankringa[1], Jace Tolleson Knee[1], Ann R Morris[1], Joseph B Wadhams[1], Sarah J Certel[2], R Steven Stowers[1]***

[1]Montana State University, Department of Microbiology and Cell Biology, Bozeman, United States; [2]The University of Montana, Division of Biological Sciences, Bozeman, United States

## eLife Assessment

This **important** study characterizes with rigorous methodology anatomical and functional aspects of the peripheral innervation of the Drosophila male reproductive tract. The **convincing** analysis reveals two distinct types of glutamatergic neurons that co-release either serotonin or octopamine. While serotonergic neurons are required for male fertility, octopaminergic neurons are dispensable. The work is providing invaluable insight into neurochemical control of insemination, peripheral motor control and neuromodulation in the male reproductive tract.

*For correspondence:
sstowers@montana.edu

Competing interest: The authors declare that no competing interests exist.

**Abstract** The essential outcome of a successful mating is the transfer of genetic material from males to females in sexually reproducing animals from insects to mammals. In males, this culminates in ejaculation, a precisely timed sequence of organ contractions driven by the concerted activity of interneurons, sensory neurons, and motor neurons. Although central command circuits that trigger copulation have been mapped, the motor architecture and the chemical logic that couple specific neuronal subclasses to organ-specific contractility, seminal fluid secretion, and sperm emission remain largely uncharted. This gap in knowledge limits our ability to explain how neural circuits adapt to varying contexts and how their failure contributes to infertility. Here, we present an in-depth anatomical and functional analysis of the motor neurons that innervate the internal male reproductive tract of *Drosophila melanogaster*. We identify two classes of multi-transmitter motor neurons based on neurotransmitter usage, namely octopamine and glutamate neurons (OGNs) and serotonin and glutamate neurons (SGNs), each with a biased pattern of innervation: SGNs predominate in the accessory glands, OGNs in the ejaculatory duct, with equal contributions of each to the seminal vesicles. Both classes co-express vesicular transporters for glutamate (vGlut) and amines (vMAT), confirming their dual chemical identity. Their target organs differentially express receptors for glutamate, octopamine, and serotonin, suggesting combinatorial neuromodulation of contractility. Functional manipulations show that SGNs are essential for male fertility but OGNs are dispensable. Glutamatergic transmission from both classes is also dispensable for fertility. These findings provide the first high-resolution map linking multi-transmitter motor neurons to specific reproductive organs, reveal an unexpected division of labor between serotonergic and octopaminergic signaling pathways, and establish a framework for dissecting conserved neural principles that govern ejaculation and male fertility.

## Introduction

Reproduction is a biological imperative of all species. For most sexually reproducing animal species, this involves locating a mate, courtship, and most essentially, the transfer of genetic material via copulation. The endpoint of copulation, ejaculation, consists of two coordinated phases, emission and expulsion (*Alwaal et al., 2015*; *Soni et al., 2022*). During the emission phase, sperm and seminal fluid are relocated from their site of origin to a central canal for transfer to the female during the subsequent expulsion phase. The series of events that culminate in ejaculation are regulated by a spinal ejaculation generator located in the lumbar L3-L4 region of the spinal cord of mammals that executes a motor program coordinating a series of muscle contractions and relaxations of the internal and external male genitalia (*Truitt and Coolen, 2002*; *Carro-Juárez et al., 2003*; *Chéhensse et al., 2017*).

While male external genitalia exhibit remarkable evolutionary variation (*Eberhard, 1985*), internal male reproductive organs exhibit remarkable evolutionary conservation over the last 600 million years since the last common ancestor of insects and mammals (*Guerrero et al., 2004*; *Gomes et al., 2012*; *Beguelini et al., 2013*; *Tomas et al., 2019*; *Fromme et al., 2021*; *Oliveira et al., 2021*; *Yartsev and Evseeva, 2021*; *da Maciel et al., 2024*). These evolutionarily conserved structures of the male reproductive system include paired testes, seminal vesicles (SVs), and semen-producing organs (prostate glands in humans, accessory glands (AGs) in insects), as well as a singular duct (urethra in humans, ejaculatory duct (ED) in insects) through which sperm and seminal fluid flow during mating.

*Drosophila melanogaster* has been instrumental in revealing fundamental principles of male reproductive behavior and physiology that extend beyond insects, including how neural circuits control mating drive, courtship, and the mechanics and duration of copulation (*Acebes et al., 2004*; *Billeter et al., 2006a*; *Villella and Hall, 2008*; *Tayler et al., 2012*; *Crickmore and Vosshall, 2013*; *Pavlou et al., 2016*; *Jois et al., 2018*; *Baker et al., 2024*). Among the findings of these studies is the existence of neural circuitry in the ventral nerve cord that regulates ejaculation similar to the spinal ejaculation generator circuitry of mammals, suggesting an ancient evolutionary origin. How the motor neurons directly innervating the internal male reproductive organs control the events leading to ejaculation is less well understood. It is also known in *Drosophila* and other species, including humans, that male

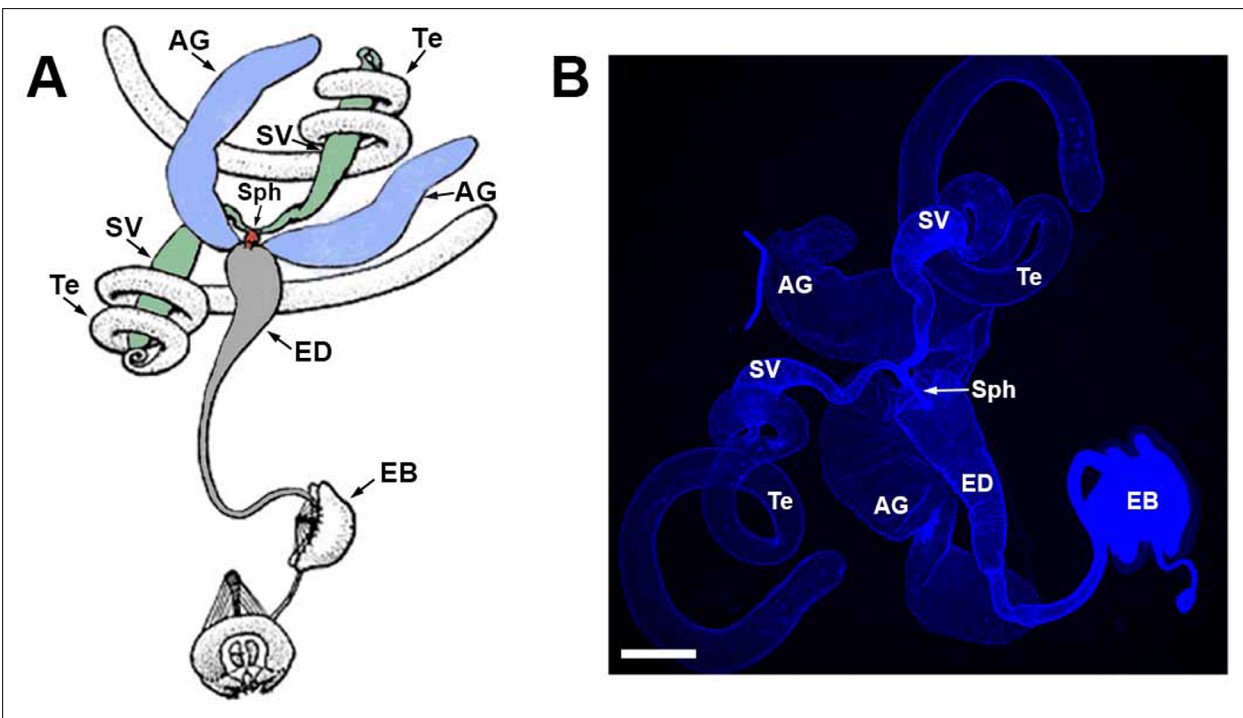

**Figure 1.** The *Drosophila* male reproductive system. (**A**) Schematic diagram showing paired testes (uncolored), SVs (green), and AGs (purple), Sph (red), ED (gray), and EB (uncolored). Modified version of Figure 38A page 508 (*Demerec, 1950*). Used with kind permission from Cold Spring Harbor Laboratory (CSHL) Press. (**B**) Actual male reproductive system. Te-testes, SV-seminal vesicle, AG-accessory gland, Sph-sphincter, ED-ejaculatory duct, EB-ejaculatory bulb. Scale bar: 200 μm.

ejaculate quality is plastic and can be adapted to environmental conditions, especially the level of male-male competition (*Gage and Baker, 1991*; *Wedell and Cook, 1999*; *Pizzari et al., 2003*; *Pound and Gage, 2004*; *Bretman et al., 2009*; *Garbaczewska et al., 2013*; *Hopkins et al., 2019*; *DeLecce et al., 2025*), but the neural mechanisms underlying these adaptations remain to be identified.

Previous studies have determined there are at least three types of motor neurons innervating the *Drosophila* internal male reproductive system distinguished by neurotransmitter usage, including serotonergic (*Lee and Hall, 2001*), octopaminergic (*Pauls et al., 2018*), and glutamatergic (*Pavlou et al., 2016*) motor neurons, although there may be at least some overlap in neurotransmitter usage as serotonin/glutamate multi-transmitter motor neurons have been reported in the seminal vesicle (*Castellanos et al., 2013*). How these neurons coordinate the motor program to elicit ejaculation is an open question, as is their anatomical organization relative to each other. Clarifying the targets and anatomy of these motor neuron classes will uncover critical principles for how neuronal coordination of internal male reproductive organ function contributes to male fertility. To address these questions, we performed a comprehensive analysis of the neurons innervating each internal organ, defined their neurotransmitter complements and receptor profiles, and tested their necessity for successful sperm transfer and fecundity.

## Results

### *Drosophila* male reproductive system anatomy

The *Drosophila* male reproductive system consists of paired testes, SVs, and AGs, as well as a singular ED and ejaculatory bulb (*Figure 1*; *Demerec, 1950*). The testes are surrounded by smooth muscle with no neuronal innervation, while the SVs, AGs, and ED are ensheathed by neuronally innervated skeletal muscles (*Lee and Hall, 2001*; *Billeter et al., 2006b*; *Susic-Jung et al., 2012*). The ejaculatory bulb also possesses neuronally innervated skeletal muscle for pumping sperm out of the ED to the female. The SVs fuse just prior to termination in the ED where they form a narrow sphincter that opens during mating to allow sperm to flow. The AGs, insect equivalent of mammalian prostate glands, produce the bulk of seminal fluid (*Wolfner, 1997*; *Majane et al., 2022*) and remain separate before terminating in the ED (*Bairati, 1968*). Underlying the muscles of the internal male reproductive system organs are a layer of secretory epithelial cells (*Bairati, 1968*) that are the cellular source of the seminal fluid components (*Sepil et al., 2019*; *Wigby et al., 2020*).

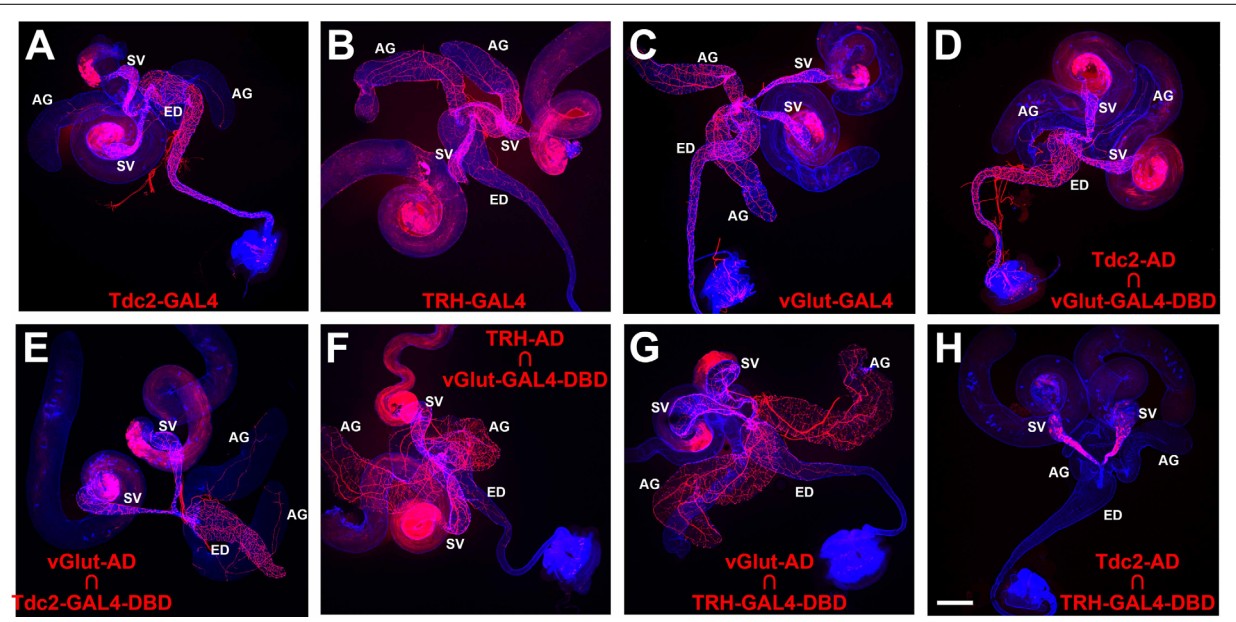

**Figure 2.** Gal4 and Split-GAL4 expression patterns in the *Drosophila* male reproductive system. (**A**) *Tdc2-GAL4*; (**B**) *TRH-GAL4*; (**C**) vGlut-GAL4; (**D**) Tdc2-AD ∩ *vGlut-GAL4-DBD*; (**E**) *vGlut-AD ∩ Tdc2-GAL4-DBD*; (**F**) *TRH-AD ∩ vGlut-GAL4-DBD*; (**G**) *vGlut-AD ∩ TRH-GAL4-DBD*; (**H**) *Tdc2-AD ∩ TRH-GAL4-DBD*. Scale bar: 200 µm.

## Neurotransmitter identity of neurons innervating internal male reproductive organs

To confirm previous reports of innervation of the *Drosophila* male reproductive system by octopaminergic, serotonergic, and glutamatergic neurons, GAL4 drivers for the octopamine (OA) neurotransmitter synthesis enzyme tyrosine decarboxylase 2 (Tdc2), the serotonin (5-Hydroxytryptamine, 5-HT) neurotransmitter synthesis enzyme tryptophan hydroxylase (TRH) and the vesicular glutamate transporter (vGlut) were used to drive expression of a *UAS-CD8-mCherry* plasma membrane reporter. As expected, all three drivers exhibited expression in neurons of the male reproductive system, but with distinguishable differences in expression patterns. *Tdc2-GAL4* exhibited dense innervation of the SV and ED with sparse innervation of the AG (*Figure 2A*). *TRH-GAL4* exhibited dense innervation of the SV and AGs, with modest innervation of the ED (*Figure 2B*), and *vGlut-GAL4* exhibited dense innervation of all three organs (*Figure 2C*).

## OA/glutamate neurons are distinct from 5-HT/glutamate neurons

To determine if the observed neuronal expression of the GAL4 drivers for Tdc2, TRH, and vGlut are coincident or exclusive, two separate approaches were implemented. In the first approach, split-GAL4 drivers specific for each neurotransmitter were tested in pairwise combinations. Both *Tdc2-AD* paired with *vGlut-GAL4-DBD* (*Figure 2D*) and *vGlut-AD* paired with *Tdc2-GAL4-DBD* (*Figure 2E*) exhibited expression patterns very similar to *Tdc2-GAL4* with dense innervation of the SV and ED, and sparse innervation of the AGs. These results suggest that all the Tdc2 neurons innervating the male reproductive system are OA/glutamate dual neurotransmitter neurons (OGNs). Similarly, both *TRH-AD* paired with *vGlut-GAL4-DBD* (*Figure 2F*) and *vGlut-AD* paired with *TRH-GAL4-DBD* (*Figure 2G*) exhibited expression patterns very similar to *TRH-GAL4* with dense innervation of the SVs and AGs with modest innervation of the ED. These results suggest that all the serotonergic neurons innervating the male reproductive system are 5-HT/glutamate dual neurotransmitter neurons (SGNs). To determine if there is overlap between the OGNs and SGNs, *Tdc2-AD* was paired with *TRH-GAL4-DBD* with the result that no neuronal expression was observed (*Figure 2H*). This result suggests the OGNs and SGNs are exclusive and non-overlapping.

A second approach to assess overlap between octopaminergic, serotonergic, and glutamatergic neurons of the male reproductive system utilized pairwise combinations of GAL4 and LexA drivers specific for each neurotransmitter. Pairing *TRH-GAL4* and *Tdc2-LexA* together with *UAS-6XGFP* and *LexAop-6XmCherry* reporters revealed no overlap in either the SVs (*Figure 3A-F*), ED (*Figure 3G-L*), or AGs (*Figure 3M-R*). These results are consistent with the split-GAL4 result above of no intersection between octopaminergic and serotonergic neurons of the male reproductive system, although both neuron types are often tightly fasciculated, especially in the SVs. Pairing *vGlut-LexA* with *TRH-GAL4* revealed a high degree of overlap in the SVs (*Figure 3—figure supplement 1A–F*), partial overlap in the ED (*Figure 3—figure supplement 1G–L*), and near complete overlap in the AGs (*Figure 3—figure supplement 1M-R*). Pairing *vGlut-LexA* with *Tdc2-GAL4* revealed significant overlap in the SVs (*Figure 3—figure supplement 2A-F*), majority overlap in the ED (*Figure 3—figure supplement 2G-L*), and limited overlap in the AGs (*Figure 3—figure supplement 2M-R*). The results of the split-GAL4 intersectional strategy and the GAL4/LexA overlap strategy are entirely consistent with each other. Taken together, they demonstrate that the *Drosophila* internal male reproductive system receives input from two distinct and non-overlapping motor-neuron populations, one population that co-expresses octopamine and glutamate (OGNs) and another distinct population that co-expresses serotonin and glutamate (SGNs). The SV receives roughly equal innervation from both classes, the AG is innervated more extensively by SGNs, and the ED is innervated more extensively by OGNs.

## OGNs are *fru +/dsx +* while SGNs are *fru+/dsx -*

Neurons that innervate the male reproductive tract are expected to be sexually dimorphic (*Billeter and Goodwin, 2004*; *Yamamoto, 2007*; *Siwicki and Kravitz, 2009*; *Verhulst and van de Zande, 2015*). To determine whether they express the canonical sex-determination genes *fruitless* (*fru*) and/or *doublesex* (*dsx*), fru-GAL4 and vGlut-LexA were paired with *UAS-6XGFP* and *LexAop-6XmCherry*. This experiment revealed complete overlap: every vGlut-positive neuron in the reproductive system—both OGNs and SGNs—was also *fru*-positive (*Figure 3—figure supplement 3A–C*). To examine *dsx* expression, the octopaminergic driver *Tdc2-AD* was combined with *dsx-GAL4-DBD*. The resulting

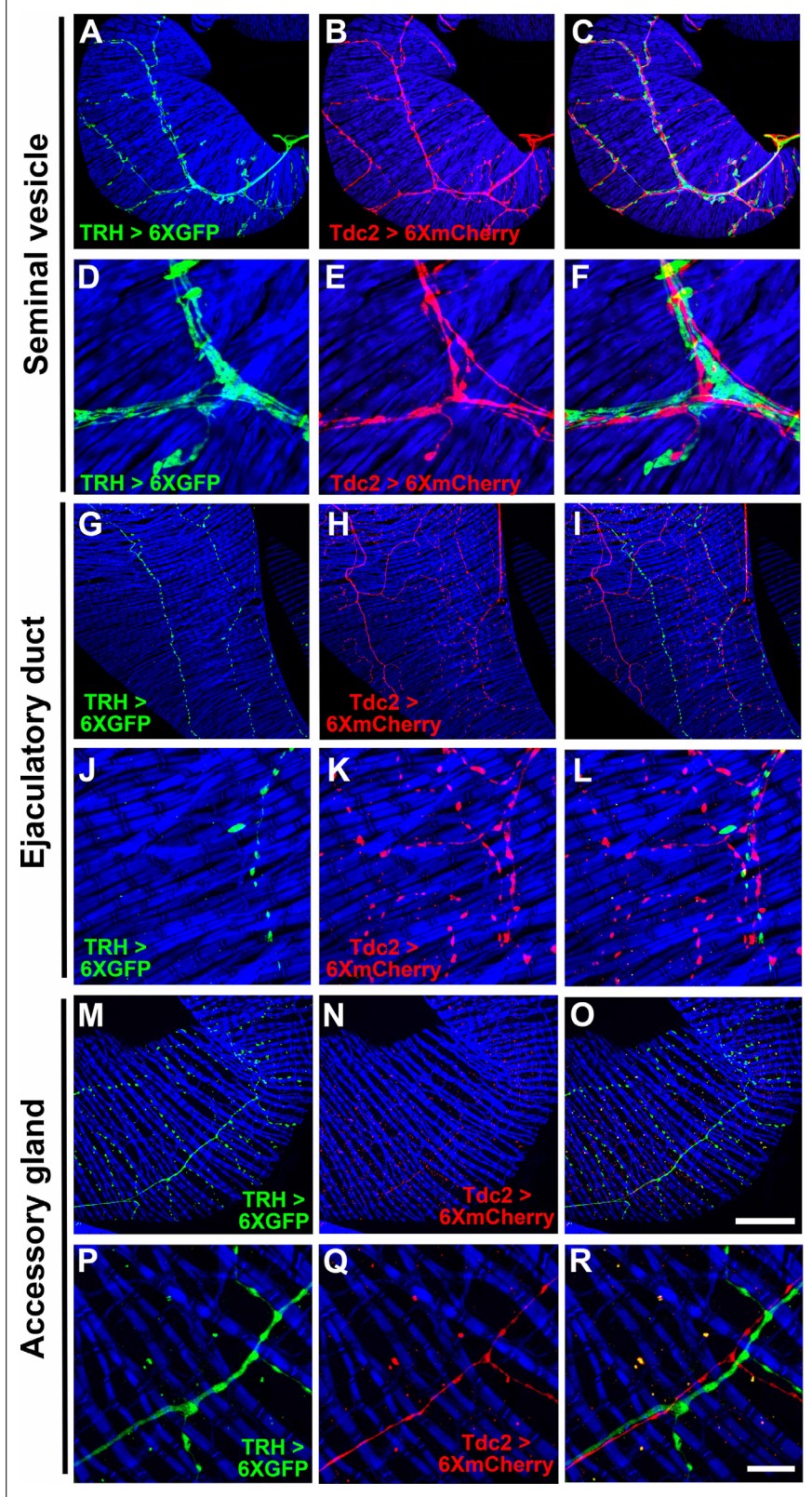

**Figure 3.** Expression of *TRH-GAL4* and *Tdc2-LexA* in the *Drosophila* male reproductive system. (**A, D, G, J, M, P**) *TRH-GAL4, UAS-6XGFP*; (**B, E, H, K, N, Q**) *Tdc2-LexA, LexAop-6XmCherry*. (**C, F, I, L, O, R**) overlay. Scale bars: O-50 μm; R-10 μm.

*Figure 3 continued on next page*

*Figure 3 continued*

The online version of this article includes the following figure supplement(s) for figure 3:

**Figure supplement 1.** Expression of *vGlut-LexA* and *TRH-GAL4* in the *Drosophila* male reproductive system.

**Figure supplement 2.** Expression of *vGlut-LexA* and *Tdc2-GAL4* in the *Drosophila* male reproductive system.

**Figure supplement 3.** Fruitless and Doublesex expression in the *Drosophila* male reproductive system.

pattern (dense terminals in the SVs and ED, sparse arbors in the AGs; *Figure 3—figure supplement 3D*) matched that of *Tdc2-GAL4* alone, indicating that all OGNs are *dsx*-positive. In contrast, pairing the serotonergic *TRH-AD* with *dsx-GAL4-DBD* produced no signal in the reproductive tract (*Figure 3—figure supplement 3E*), showing that SGNs are *dsx*-negative. Together, these experiments establish that OGNs are *fru+/dsx+*, whereas SGNs are *fru+/dsx–*, underscoring a molecular distinction between the two neuromodulatory inputs to the male reproductive system.

## Non-overlapping expression of TβH and Tdc2 with 5-HT confirms exclusivity of OGNs and SGNs

Several predictions follow from the finding that non-overlapping sets of OGNs and SGNs innervate the male reproductive system. Among these are that the neurotransmitter synthesis enzymes for OA, Tyramine-β-hydroxylase (TβH), and Tyrosine decarboxylase 2 (Tdc2) should be exclusively expressed in the OGNs along with the sole *Drosophila* vesicular glutamate transporter (vGlut), while 5-HT should be exclusively expressed in the SGNs along with vGlut-40XV5. To test these predictions, vGlut-40XV5 expression was separately assessed in combination with TβH or Tdc2. 5-HT expression, visualized with an anti-5-HT antibody, was also included in this analysis to distinguish SGNs. As predicted, extensive vGlut-40XV5 expression was observed throughout the male reproductive system except for the testes (*Figure 4A and G*) with a sharp boundary of neuronal innervation at the testicular duct (arrows, *Figure 4A*). Also, as predicted, TβH-GFP and Tdc2 were observed in the male reproductive system with expression patterns similar to that of *Tdc2-GAL4* (*Figure 4B and H*, respectively). Pairwise combinations of overlapping expression of vGlut-40XV5, 5-HT, and either TβH (*Figure 4D-F*) or Tdc2 (*Figure 4J-L*) are also shown.

Higher resolution images of vGlut-40XV5, TβH-GFP, Tdc2, and 5-HT in the male reproductive system allow an assessment of overlapping expression. In the SV, broad robust expression was observed for vGlut-40XV5 (*Figure 4—figure supplement 1A, G, M, and S*) and 5-HT (*Figure 4—figure supplement 1C, I, O, and U*), with somewhat more restricted expression for TβH-GFP (*Figure 4—figure supplement 1B and H*) and Tdc2 (*Figure 4—figure supplement 1N, T*). Pairwise overlapping expression is also shown for vGlut-40XV5 and 5-HT with TβH-GFP (*Figure 4—figure supplement 1D-F and J-L*) and Tdc2 (*Figure 4—figure supplement 1P-R and V-X*). In the highest resolution images, coincident expression is evident between vGlut-40XV5 with TβH-GFP (*Figure 4—figure supplement 1J*) and vGlut-40XV5 with Tdc2 (*Figure 4—figure supplement 1V*). Consistent with results above, exclusivity of expression is apparent between TβH-GFP and 5-HT (*Figure 4—figure supplement 1L*) and between Tdc2 and 5-HT (*Figure 4—figure supplement 1X*), although the tight fasciculation of OGNs and SGNs is evident.

Similar results were obtained in higher resolution images of the ED and AGs, although the extent of innervation of the different amines varies between these tissues with more OGNs than SGNs in the ED and the reciprocal in the AGs. In higher resolution images of the ED, strong extensive expression was observed for vGlut-40XV5 (*Figure 4—figure supplement 2A, G, M, and S*) with moderately less expression for TβH-GFP (*Figure 4—figure supplement 2B and H*) and Tdc2 (*Figure 4—figure supplement 2N, T*), with noticeably less expression of 5-HT (*Figure 4—figure supplement 2C, I, O, and U*). Pairwise overlapping expression is also shown for vGlut-40XV5 and 5-HT with TβH-GFP (*Figure 4—figure supplement 2D-F and J-L*) and Tdc2 (*Figure 4—figure supplement 2P-R and V-X*). In the highest resolution images, coincident expression is evident between vGlut-40XV5 with TβH-GFP (*Figure 4—figure supplement 2J*) and vGlut-40XV5 with Tdc2 (*Figure 4—figure supplement 2V*). As in the SV, exclusivity of expression is apparent between TβH-GFP and 5-HT (*Figure 4—figure supplement 2L*) and between Tdc2 and 5-HT (*Figure 4—figure supplement 2X*), also with tight fasciculation of the OGNs and SGNs.

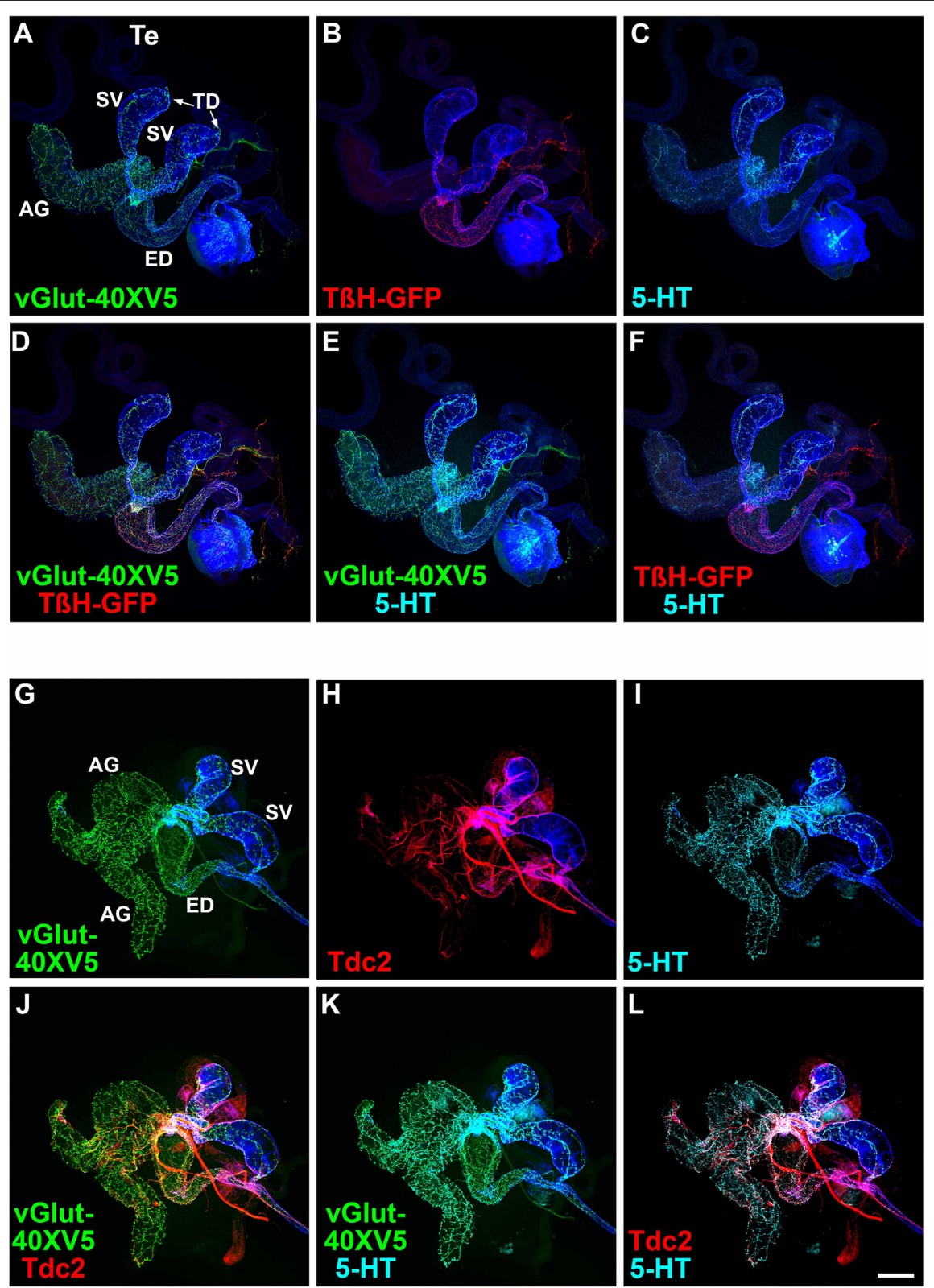

**Figure 4.** Expression of vGlut, TβH-GFP, Tdc2, and 5-HT in the *Drosophila* male reproductive system. (**A**) vGlut-40XV5; (**B**) TβH-GFP; (**C**) 5-HT; (**D**) vGlut-40XV5, TβH-GFP overlay; (**E**) vGlut-40XV5, 5-HT overlay; (**F**) TβH-GFP, 5-HT overlay; (**G**) vGlut-40XV5; (**H**) Tdc2; (**I**) 5-HT; (**J**) vGlut-40XV5, Tdc2 overlay; (**K**) vGlut-40XV5, 5-HT overlay; (**L**) Tdc2, 5-HT overlay. Scale bar: 200 μm.

*Figure 4 continued on next page*

*Figure 4 continued*

The online version of this article includes the following figure supplement(s) for figure 4:

**Figure supplement 1.** Expression of vGlut, TβH-GFP, Tdc2, and 5-HT in the seminal vesicle (SV) of the *Drosophila* male reproductive system.

**Figure supplement 2.** Expression of vGlut, TβH-GFP, Tdc2, and 5-HT in the ejaculatory duct (ED) of the *Drosophila* male reproductive system.

**Figure supplement 3.** Expression of vGlut, TβH-GFP, Tdc2, and 5-HT in the accessory gland (AG) of the *Drosophila* male reproductive system.

**Figure supplement 4.** Expression of vGlut, TβH-GFP, and 5-HT at the junction of the seminal vesicle (SV) and accessory glands (AGs) with the ejaculatory duct (ED) of the *Drosophila* male reproductive system.

In higher resolution images of the AGs, robust widespread expression was observed for vGlut-40XV5 (*Figure 4—figure supplement 3A, G, M, and S*) and 5-HT (*Figure 4—figure supplement 3C, I, O, and U*), with no expression of TβH-GFP (*Figure 4—figure supplement 3B and H*), and sparse expression of Tdc2 (*Figure 4—figure supplement 3N, T*). Pairwise overlapping expression is also shown for vGlut-40XV5 and 5-HT with TβH-GFP (*Figure 4—figure supplement 3D-F and J-L*) and Tdc2 (*Figure 4—figure supplement 3P-R and V-X*). In the highest resolution images, coincident expression is evident between vGlut-40XV5 and Tdc2 (*Figure 4—figure supplement 3V*). As in the SV and ED, exclusivity of expression is apparent between Tdc2 and 5-HT (*Figure 4—figure supplement 3X*).

The expression of vGlut-40XV5, TβH-GFP, and 5-HT was also assessed at the junction of the SVs and AGs with the ED. vGlut-40XV5 innervation of both the SV and AG is particularly dense (*Figure 4—figure supplement 4A*). Significant expression of TβH-GFP and 5-HT was also observed (*Figure 4—figure supplement 4B and C*, respectively). Pairwise overlapping expression for vGlut-40XV5, TβH-GFP, and 5-HT is also shown (*Figure 4—figure supplement 4D-F*). These experiments showing no overlap in expression of TβH-GFP and Tdc2 with 5-HT provide additional support to the exclusivity of OGNs and SGNs. Collectively, these findings confirm that octopaminergic vGlut-positive and serotonergic vGlut-positive motor neurons form distinct, non-overlapping populations that differentially innervate the *Drosophila* male reproductive tract, underscoring strict transmitter exclusivity within this circuit.

## Expression of vGlut and vMAT in both OGNs and SGNs further confirms multi-neurotransmitter usage

Co-labeling of vGlut-40XV5 with TβH/Tdc2 in OGNs and with 5-HT in SGNs implies both neuron types are multi-transmitter and predicts they will express not only vGlut but also the sole *Drosophila* monoamine vesicular transporter vMAT known to package both OA and 5-HT (*Deshpande et al., 2020*). Since OGNs and SGNs often fasciculate tightly together, co-conditional expression of vGlut and vMAT was implemented to visualize expression separately in each population. This was achieved using either the *TRH-GAL4* or *Tdc2-GAL4* drivers in combination with the *B2RT-STOP-B2RT-vGlut-40XMYC*, *RSRT-STOP-RSRT-6XV5-vMAT*, *UAS-B2,* and *UAS-R* recombinase transgenes. Using the *TRH-GAL4* driver to restrict conditional expression to SGNs, as predicted, both vGlut-40XMYC (*Figure 5A, E*) and 6XV5-vMAT (*Figure 5B, F*) were detected in the SV. Tdc2 expression was also included in this experiment (*Figure 5C and G*) as was a *UAS-CD8-mCherry* reporter to outline SGNs (*Figure 5D and H*). A high degree of overlap was observed between vGlut-40XMYC and 6XV5-vMAT (*Figure 5I*) with no overlap between Tdc2 and the other markers (*Figure 5J-L*).

In a reciprocal experiment in the SV using the *Tdc2-GAL4* driver and 5-HT immunostaining, both vGlut-40XMYC (*Figure 5M and Q*) and 6XV5-vMAT (*Figure 5N and R*) were detected with a mostly overlapping distribution (*Figure 5U*). 5-HT immunostaining was in close proximity but did not overlap with any of the other markers (*Figure 5V-X*). It is also notable that the morphology of the SGNs is distinct from the OGNs with the SGNs being larger and more extensively branched compared to the OGNs (SGNs-*Figure 5D and H*; OGNs-*Figure 5P and T*), suggesting SGNs may assert more regulatory control of SV muscle activity than OGNs.

Similar results were obtained in these co-conditional experiments for the ED and AGs. With the *TRH-GAL4* experiment in the ED, vGlut-40XMYC (*Figure 5—figure supplement 1A and E*) and 6XV5-vMAT (*Figure 5—figure supplement 1B and F*) were both present with a highly overlapping distribution (*Figure 5—figure supplement 1I*). Tdc2 did not overlap with any of the other markers (*Figure 5—figure supplement 1J-L*). With the *Tdc2-GAL4* experiment in the ED, vGlut-40XMYC

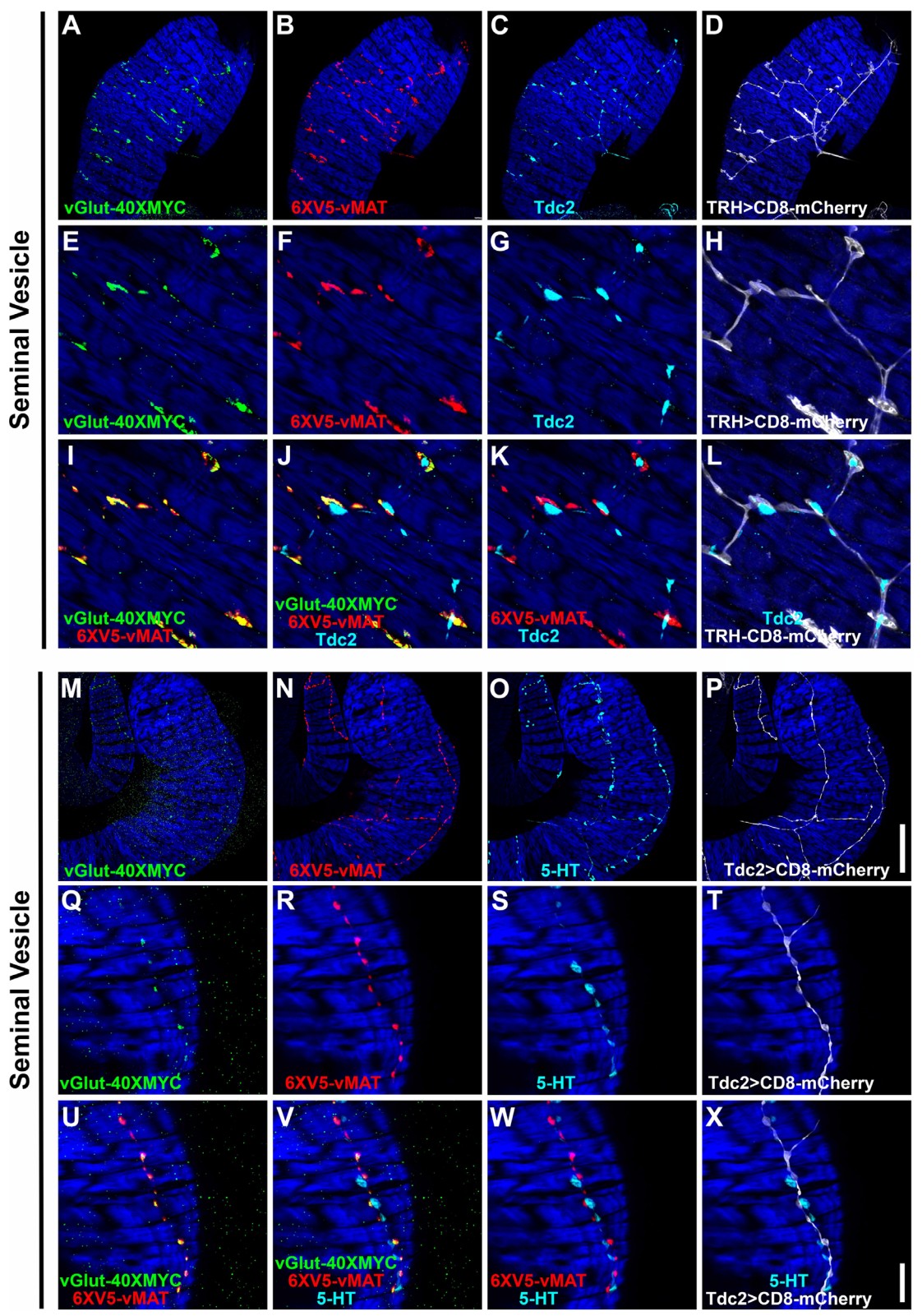

**Figure 5.** Co-conditional expression of vGlut-40XMYC and 6XV5-vMAT in the seminal vesicle (SV) of serotonergic (TRH) or octopaminergic (Tdc2) neurons of the *Drosophila* male reproductive system. (**A–D**) 63X. (**A**) vGlut-40XMYC; (**B**) 6XV5-vMAT; (**C**) Tdc2; (**D**) TRH >CD8-mCherry. (**E–L**) 63X zoom 4X. (**E**) vGlut-40XMYC; (**F**) 6XV5-vMAT; (**G**) Tdc2; (**H**) TRH >CD8-mCherry; (**I**) vGlut-40XMYC, 6XV5-vMAT overlay; (**J**) vGlut-40XMYC, 6XV5-vMAT, Tdc2 overlay; (**K**) 6XV5-vMAT, Tdc2 overlay; (**L**) Tdc2, TRH >CD8-mCherry overlay. (**M–P**) 63X. (**M**) vGlut-40XMYC; (**N**) 6XV5-vMAT; (**O**) 5-HT; (**P**) Tdc2>CD8-

*Figure 5 continued*

mCherry. (**Q–X**) 63X zoom 4X. (**Q**) vGlut-40XMYC; (**R**) 6XV5-vMAT; (**S**) 5-HT; (**T**) Tdc2>CD8-mCherry; (**U**) vGlut-40XMYC, 6XV5-vMAT overlay; (**V**) vGlut-40XMYC, 6XV5-vMAT, 5-HT overlay; (**W**) 6XV5-vMAT, 5-HT overlay; (**X**) 5-HT, Tdc2>CD8-mCherry overlay. Scale bars: P-50 µm; X-10 µm.

The online version of this article includes the following figure supplement(s) for figure 5:

**Figure supplement 1.** Co-conditional expression of vGlut-40XMYC and 6XV5-vMAT in the ejaculatory duct (ED) of serotonergic (TRH) or octopaminergic (Tdc2) neurons of the *Drosophila* male reproductive system.

**Figure supplement 2.** Co-conditional expression of vGlut-40XMYC and 6XV5-vMAT in the accessory gland (AG) of serotonergic (TRH) or octopaminergic (Tdc2) neurons of the *Drosophila* male reproductive system.

**Figure supplement 3.** Co-conditional expression of vGlut-40XMYC and 6XV5-vMAT in serotonergic neurons of the *Drosophila* ejaculatory duct (ED) using expansion microscopy.

**Figure supplement 4.** Co-conditional expression of vGlut-40XMYC and 6XV5-vMAT in octopaminergic neurons of the *Drosophila* ejaculatory duct (ED) using expansion microscopy.

**Figure supplement 5.** Nuclei of unexpanded and expanded ejaculatory ducts (EDs).

(*Figure 5—figure supplement 1M and Q*) and 6XV5-vMAT (*Figure 5—figure supplement 1N and R*) were both observed in a highly overlapping distribution (*Figure 5—figure supplement 1U*). 5-HT immunostaining did not overlap with any of the other markers (*Figure 5—figure supplement 1V-X*). Note that morphological differences between the SGNs and OGNs in the ED are not as easily distinguished as in the SV.

With the *TRH-GAL4* experiment in the AGs, both vGlut-40XMYC (*Figure 5—figure supplement 2A and E*) and 6XV5-vMAT (*Figure 5—figure supplement 2B and F*) are present in a largely overlapping distribution (*Figure 5—figure supplement 2I*). As with the SV and ED, Tdc2 expression did not overlap with any of the other markers (*Figure 5—figure supplement 2J-L*). With the *Tdc2-GAL4* experiment in the AG, both vGlut-40XMYC (*Figure 5—figure supplement 2M and Q*) and 6XV5-vMAT (*Figure 5—figure supplement 2N and R*) are present in an overlapping configuration (*Figure 5—figure supplement 2U*). The distribution of 5-HT is distinct from any of the other markers (*Figure 5—figure supplement 2V-X*). In the AGs, the morphology of the SGNs (*Figure 5—figure supplement 2D and H*) is distinct from that of the OGNs (*Figure 5—figure supplement 2P and T*).

Since OGNs and SGNs are intermingled in tightly fasciculated neuron bundles, we used intersectional labeling to visualize the glutamate transporter vGlut-40XV5 and the monoamine transporter 6XV5-vMAT in each class separately. Together, the results show that both vesicular transporters are expressed in OGNs and SGNs in an overlapping distribution while the OGNs and SGNs remain anatomically distinct throughout the male reproductive tract.

## Expansion microscopy reveals prominent co-occurrence of vGlut and vMAT on ED synaptic vesicles of both OGNs and SGNs

Although there is apparent overlap between vGlut-40XMYC and 6XV5-vMAT in both SGNs and OGNs, it is not possible to resolve whether they reside on the same or different synaptic vesicles using standard microscopy methods. This constraint is due to the ~250 nm diffraction limit of light at visible wavelengths (*Abbe, 1873*). To overcome this limitation, expansion microscopy was performed on EDs using co-conditional expression of vGlut-40XMYC and 6XV5-vMAT separately in SGNs and OGNs as above. The protocol utilized in these experiments reported 10X expansion in a single round (*Damstra et al., 2023*), and this agreed well with our empirically determined expansion factor of 10.47X based on the size comparison of nuclei area in muscles of the unexpanded and expanded EDs (*Figure 5—figure supplement 5A and B*, respectively). In a representative example of expanded SGNs, 6XV5-vMAT (*Figure 5—figure supplement 3A*) and vGlut-40XMYC (*Figure 5—figure supplement 3B*) revealed a significantly overlapping distribution (*Figure 5—figure supplement 3C*). To quantify the co-occurrence of vGlut-40XMYC and 6XV5-vMAT, the BIOP JACop ImageJ plug-in was used to calculate a Mander's coefficient, a metric that calculates the fraction of one signal that co-occurs with another. The output of the BIOP JACop plug-in revealed substantial co-occurrence of 6XV5-vMAT and vGlut-40XMYC (*Figure 5—figure supplement 3D-I*), including a graphical representation (*Figure 5—figure supplement 3J*). For the SGNs, the numerical Mander's coefficients for 6XV5-vMAT co-occurrence with vGlut-40XMYC were 67.9% and vGlut-40XMYC co-occurrence with 6XV5-vMAT were 57.2% (*Figure 5—figure supplement 3K*).

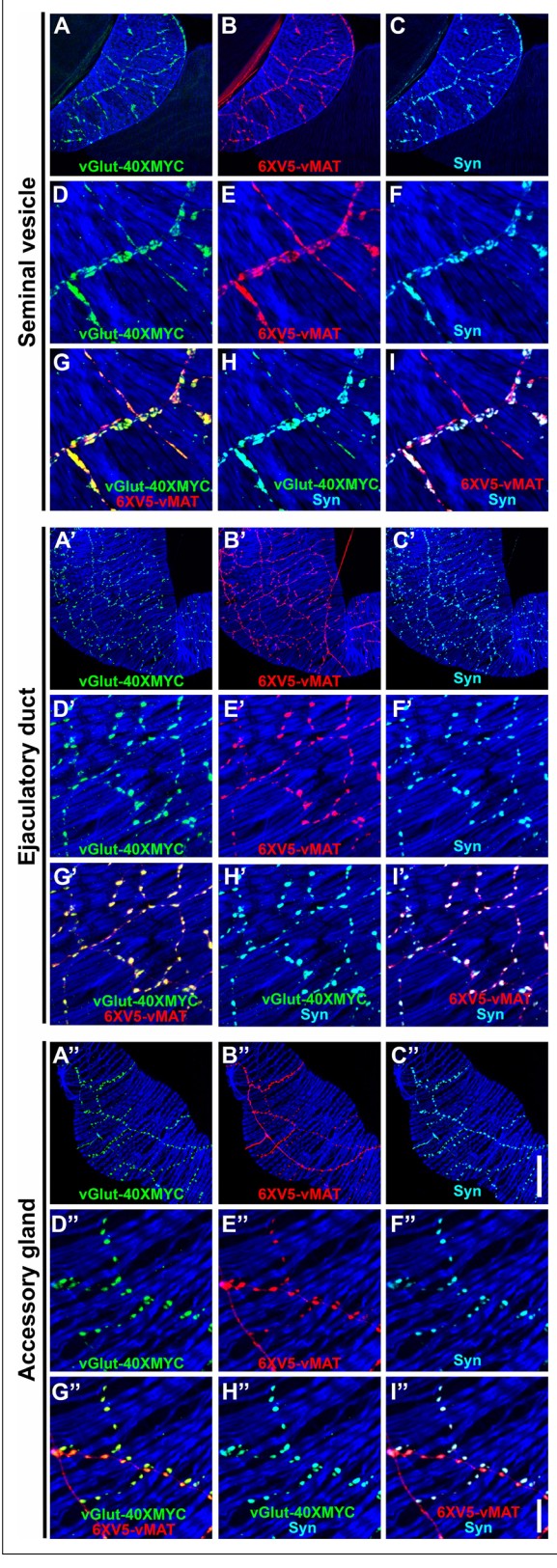

**Figure 6.** Expression of the synaptic vesicle marker Synapsin in combination with vGlut-40XMYC and 6XV5-vMAT in the seminal vesicle (SV), ejaculatory duct (ED), and accessory gland (AG) of the *Drosophila* male reproductive system. (**A–I**) SV. (**A**) vGlut-40XMYC; (**B**) 6XV5-vMAT; (**C**) Synapsin; (**D**) vGlut-40XMYC; (**E**) 6XV5-vMAT; (**F**) Synapsin; (**G**) vGlut-40XMYC, 6XV5-vMAT overlay; (**H**) vGlut-40XMYC, Synapsin overlay; (**I**) 6XV5-vMAT, Synapsin overlay.

*Figure 6 continued on next page*

*Figure 6 continued*

(**A'-I'**) ED. (**A'**) vGlut-40XMYC; (**B'**) 6XV5-vMAT; (**C'**) Synapsin; (**D'**) vGlut-40XMYC; (**E'**) 6XV5-vMAT; (**F'**) Synapsin; (**G'**) vGlut-40XMYC, 6XV5-vMAT overlay; (**H'**) vGlut-40XMYC, Synapsin overlay; (**I'**) 6XV5-vMAT, Synapsin overlay. (**A"-I"**) AG. (**A"**) vGlut-40XMYC; (**B"**) 6XV5-vMAT; (**C"**) Synapsin; (**D"**) vGlut-40XMYC; (**E"**) 6XV5-vMAT; (**F"**) Synapsin; (**G"**) vGlut-40XMYC, 6XV5-vMAT overlay; (**H"**) vGlut-40XMYC, Synapsin overlay; (**I"**) 6XV5-vMAT, Synapsin overlay. Scale bars: C"–50 μm; I"–10 μm.

The online version of this article includes the following figure supplement(s) for figure 6:

**Figure supplement 1.** Expression of the active zone marker Brp in combination with vGlut-40XMYC and 6XV5-vMAT in the seminal vesicle (SV), ejaculatory duct (ED), and accessory gland (AG) of the *Drosophila* male reproductive system.

**Figure supplement 2.** Expression of the post-synaptic density marker Dlg in combination with vGlut-40XMYC and 6XV5-vMAT in the seminal vesicle (SV), ejaculatory duct (ED), and accessory gland (AG) of the *Drosophila* male reproductive system.

**Figure supplement 3.** Expression of the large core dense vesicle marker IA2-GFP in combination with vGlut-40XMYC and 6XV5-vMAT in the seminal vesicle (SV) of the *Drosophila* male reproductive system.

**Figure supplement 4.** Expression of the large core dense vesicle marker IA2-GFP in combination with vGlut-40XMYC and 6XV5-vMAT in the ejaculatory duct (ED) and accessory gland s (AGs) of the *Drosophila* male reproductive system.

**Figure supplement 5.** Expression of vGlut-40XMYC, 6XV5-vMAT, 7XMYC-vAChT, and 9XV5-vGAT in the seminal vesicle (SV), ejaculatory duct (ED), and accessory gland (AG) of the *Drosophila* male reproductive system.

Expanded OGNs yielded similar results. In a representative example, 6XV5-vMAT (*Figure 5—figure supplement 4A*) and vGlut-40XMYC (*Figure 5—figure supplement 4B*) exhibited substantial overlap (*Figure 5—figure supplement 4C*). The output of the BIOP JACop plug-in also revealed majority co-occurrence in image (*Figure 5—figure supplement 4D-I*) and graphical form (*Figure 5—figure supplement 4J*). For the OGNs, the numerical values of the Mander's coefficients were 67.4% for 6XV5-vMAT co-occurrence with vGlut-40XMYC and 75.6% for vGlut-40XMYC co-occurrence with 6XV5vMAT (*Figure 5—figure supplement 4K*).

The results of the expansion microscopy experiments indicate that in both SGNs and OGNs, the majority of ED synaptic vesicles are positive for both vGlut-40XMYC and 6XV5-vMAT, and thereby they co-release glutamate and either OA (OGNs) or 5-HT (SGNs). This demonstration of a neurotransmitter co-release mechanism may be a conserved strategy for synchronizing rapid gland and muscle activity during copulation that has general implications across species for how co-transmitter neurons mediate complex motor behaviors.

## OGNs and SGNs express synaptic markers common to other neuromuscular junctions

In addition to providing information on whether a given synapse is primarily excitatory or modulatory, identifying vesicular transporters and synaptic markers allows mapping of active release sites to determine how pre- and post-synaptic specializations align. Such information is essential for (i) elucidating how each neurotransmitter influences muscle contractility and secretory function, and (ii) predicting how synaptic efficacy might change during prolonged copulation or in response to neuromodulatory feedback. To further characterize the synapses of the neurons innervating the male reproductive system, immunostaining was performed with vGlut-40XMYC, 6XV5-vMAT, and either of three well-established *Drosophila* synaptic markers, including Synapsin (Syn)-synaptic vesicles, Bruchpilot (Brp)-active zones, and Discs-large (Dlg-Drosophila homolog of mammalian PSD-95)-post-synaptic density. In the SV, strong Syn staining was observed (*Figure 6C and F*) that exhibited extensive co-localization with vGlut-40XMYC (*Figure 6H*) and 6XV5-vMAT (*Figure 6I*). Similar results were obtained in the ED and AGs. Abundant Syn expression was observed (*Figure 6C' and F'*) that exhibited near-precise overlap with vGlut-40XMYC (*Figure 6H'*) and 6XV5-vMAT (*Figure 6H and I'*). For the AG, robust Syn expression was observed (*Figure 6A, C" and F"*) that co-localized strongly with vGlut-40XMYC (*Figure 6A and H"*) and 6XV5-vMAT (*Figure 6A and I"*). It is notable that a visibly higher density of Syn expression was observed in the SV (*Figure 6A and F*) compared with the AGs (*Figure 6A and F'*) and ED (*Figure 6AF"*), presumably indicating greater neurotransmitter output.

The active zone marker Brp exhibited strong expression in the SV (*Figure 6—figure supplement 1C and G*) that overlapped almost entirely with 6XV5-vMAT. Brp expression in the ED and AG was noticeably sparser as compared to the SV. The sparse Brp expression in the ED (*Figure 6—figure supplement 1K and O*) overlapped with 6XV5-vMAT. Similarly, in the AGs, Brp expression was sparse and overlapped with 6XV5-vMAT (*Figure 6—figure supplement 1S and W*).

The post-synaptic density marker Dlg was expressed broadly throughout the male reproductive system (*Figure 6—figure supplement 2C*). In the SV, the bulk of Dlg expression is predominantly in the epithelial cells with lower levels of expression in the muscles (*Figure 6—figure supplement 2G and K*) and no preferential post-synaptic accumulation (*Figure 6—figure supplement 2H and L*). Similar results were observed in the ED and AGs. In the ED at the level of the muscle, Dlg expression is diffuse (*Figure 6—figure supplement 2O*) and does not accumulate post-synaptically opposite synapses in the muscles (*Figure 6—figure supplement 2P*). In the AGs, Dlg expression is also diffuse in the muscles (*Figure 6—figure supplement 2T*) and does not accumulate post-synaptically opposite synapses in the muscle (*Figure 6C and U*). Images of Dlg expression in the epithelial layer of the ED (*Figure 6C and Q*) and AG (*Figure 6C and V*) reveal apparent membrane localization in a honeycomb pattern. The lack of post-synaptic accumulation of Dlg opposite the synapses of SGNs and OGNs in the male reproductive system is reminiscent of larval type II OGNs and distinct from larval type I glutamatergic neurons that exhibit prominent post-synaptic concentration of Dlg (*Parnas et al., 2001*). It is also notable that there is no vGlut-40XMYC that does not co-localize with 6XV5-vMAT, indicating there are no neurons innervating the SVs, AGs, or ED that use glutamate as their sole small molecule neurotransmitter. The absence of synapses with post-synaptic Dlg accumulation, a characteristic of type I glutamate-only larval neurons, also supports this assertion. Defining the molecular architecture of these synapses is critical for understanding how they package, release, and receive neurotransmitters to shape the strength and specificity of communication between motor neurons and target organs.

## LDCVs are present in OGNs and SGNs

In addition to synaptic transmission via synaptic vesicles, neurons are also known to communicate using neuropeptides released via Large Dense Core Vesicles (LDCVs). To determine whether neurons innervating the male reproductive system contain LDCVs, immunostaining was performed with the LDCV marker IA2-GFP (*Yu et al., 2025*) in combination with vGlut-40XMYC and 6XV5-vMAT. Abundant IA2-GFP expression was observed throughout the male reproductive system (*Figure 6—figure supplement 3C*). In the SVs, IA2-GFP (*Figure 6—figure supplement 3F and L*) localizes to the same pre-synaptic regions of the neurons as vGlut-40XMYC (*Figure 6—figure supplement 3D and J*) and 6XV5-vMAT (*Figure 6—figure supplement 3E and K*). In the highest resolution images, there appears to be some overlap between IA2-GFP and vGlut-40XMYC (*Figure 6—figure supplement 3N*) as well as between IA2-GFP and 6XV5-vMAT (*Figure 6—figure supplement 3O*), but a substantial fraction of IA2-GFP distributes in close proximity to, but is distinct from, vGlut-40XMYC and 6XV5-vMAT. This distribution pattern is not entirely surprising as it is consistent with the prior observation that vesicular neurotransmitter transporters have been detected on LDCVs (*Yu et al., 2025*).

Similar results were observed in the ED and AGs. In the ED, prominent IA2-GFP expression was evident (*Figure 6—figure supplement 4C and I*) that partially co-localized with vGlut-40XMYC (*Figure 6—figure supplement 4E and K*) and 6XV5-vMAT (*Figure 6—figure supplement 4F and L*). Considerable IA2-GFP expression was also apparent in the AGs (*Figure 6—figure supplement 4O and U*) that partially overlapped with vGlut-40XMYC (*Figure 6—figure supplement 4Q and W*) and 6XV5-vMAT (*Figure 6—figure supplement 4R and X*). This finding of pronounced expression of IA2-GFP throughout the pre-synaptic terminals of neurons innervating the male reproductive system suggests substantial LDCVs are present in both SGNs and OGNs, and that these neurons are thus communicating not just via small molecule neurotransmitters inside synaptic vesicles but also by neuropeptides contained within LDCVs.

## No cholinergic or GABAergic transmission at OGN or SGN synapses

Although to our knowledge, there have been no motor neurons reported in *Drosophila* that use acetylcholine or GABA as a neurotransmitter, to rule out these possibilities for the male reproductive system, *Drosophila* containing genome-edited 7XMYC-vAChT and 9XV5-vGAT were immunostained

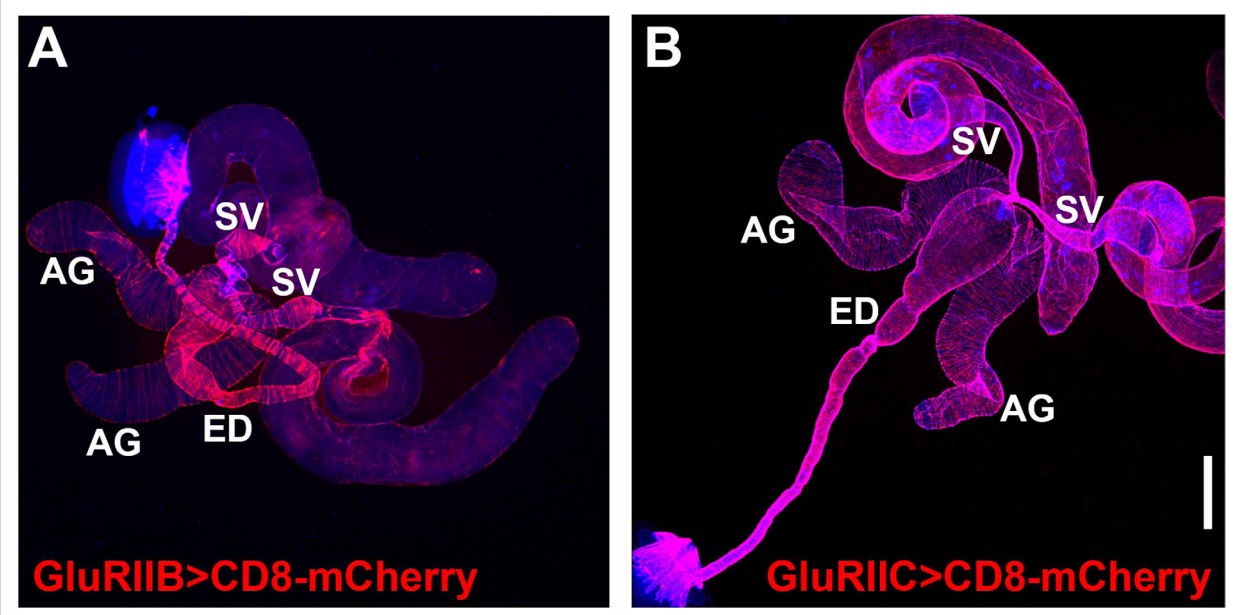

**Figure 7.** Glutamate receptor GAL4 expression patterns in *Drosophila* male reproductive system. (**A**) GluRIIB; (**B**) GluRIIC. Scale bar: 200 μm.

The online version of this article includes the following figure supplement(s) for figure 7:

**Figure supplement 1.** Octopamine (OA) receptor GAL4 expression patterns in *Drosophila* male reproductive system.

**Figure supplement 2.** 5-HT receptor GAL4 expression patterns in *Drosophila* male reproductive system.

for their corresponding epitope tags. Not unexpectedly, no expression of 7XMYC-vAChT was observed in the SVs, ED, or AGs (*Figure 6—figure supplement 5 C, G, and K*, respectively). Similarly, no expression of 9XV5-vGAT was observed in the SVs, ED, or AGs (*Figure 6—figure supplement 5D, H, and L*, respectively), indicating that neither acetylcholine nor GABA are used as neurotransmitters in the OGNs and SGNs of the *Drosophila* internal male reproductive system.

## GAL4 drivers for OA, 5-HT, and glutamate receptors express in the muscles of male reproductive organs

Given that OGNs and SGNs innervate the male reproductive system, an obvious prediction that follows is that neurotransmitter receptors for glutamate, OA, and 5-HT should be expressed in the muscles of the male reproductive system. As a first step in testing this prediction, GAL4 drivers for each of these neurotransmitters were paired with the *UAS-CD8-mCherry* reporter and assessed for expression. As it is well established that the GluRIIA-E/kainate type glutamate receptors mediate synaptic transmission at *Drosophila* NMJs (*DiAntonio, 2006*; *He and Dickman, 2025*), GAL4 drivers for all five were assessed. GAL4 drivers for GluRIIA, GluRIID, and GluRIIE showed no expression in the male reproductive system, but they also exhibited no expression in other adult or larval muscles where they are known to be expressed. These results suggest that the GAL4 drivers for these three GluRII receptors are simply non-functional and are, therefore, non-informative. However, GluRIIB-GAL4 and GluRIIC-GAL4 exhibited broad expression in the musculature of the male reproductive system (*Figure 7A and B*, respectively). This suggests glutamate is used for signaling throughout the male reproductive system, consistent with the widespread expression of vGlut in pre-synaptic terminals.

OA receptor GAL4 drivers for OAMB, OAα2R, Oct-TyrR, OAβ1R, OAβ2R, and OAβ3R were assessed for expression in the male reproductive system. OAMB-GAL4 exhibited expression in the muscles of the SVs and ED, as well as epithelial cells of the AGs (*Figure 7—figure supplement 1A*). OAα2R-GAL4 displayed strong expression in the muscles of the SVs, weak expression in the muscles of the AGs, and no expression in the ED (*Figure 7—figure supplement 1B*). Oct-TyrR-GAL4 showed strong expression in non-muscle cells of the distal AG, muscle expression in the AGs, ED, and ejaculatory bulb, expression in the testes, but no expression in the muscles of the SVs (*Figure 7—figure supplement 1C*). OAβ2R-GAL4 exhibited strong expression in the epithelial cells of the ED, muscle

expression in the SVs, ED, and ejaculatory bulb, expression in the testes, but no muscle expression in the AGs (*Figure 7—figure supplement 1E*). OAβ1R-GAL4 and OAβ3R-GAL4 exhibited broad expression in the neurons innervating the male reproductive system (*Figure 7—figure supplement 1D and F*, respectively) but no muscle expression, suggesting they are not mediating OA signaling in the muscles. OAβ3R-GAL4 also displayed expression in the testes. These results suggest OA receptors play a prominent role in the functioning of the male reproductive system and differentially mediate OA signaling in specific parts of the male reproductive system based on their restricted, non-uniform expression patterns.

5-HT receptor GAL4 drivers for 5-HT1A, 5-HT2A, 5-HT-1B, 5-HT2B, and 5-HT7 were also assessed for expression in the male reproductive system. 5-HT1A-GAL4, 5-HT1B-GAL4, and 5-HT2B-GAL4 exhibited broad expression in the neurons innervating the male reproductive system, but no muscle expression (*Figure 7—figure supplement 2A, C, and D*, respectively), suggesting these receptors are not mediating 5-HT signaling in the muscles. 5-HT2A exhibited expression specifically in the epithelial cells of the SVs (*Figure 7—figure supplement 2B*). 5-HT7 showed substantial expression in neurons, but most strikingly, strong expression in the muscles of the distal SVs, including the sphincter region, with a discrete boundary of expression (arrows, *Figure 7—figure supplement 2E*).

## GluRIIA is expressed in the SVs and AGs, but not the ED

While informative for determining cellular expression, GAL4-driven reporters for genes of interest do not provide critical mechanistic information about the subcellular distribution of the corresponding proteins. To gain further insight into the role of OA, 5-HT, and glutamate receptors in the function of the male reproductive system, the expression patterns of several receptors were assessed using epitope-tagging. To determine the subcellular distribution of the ionotropic GluRII/kainate-type glutamate receptors known to mediate fast excitatory glutamatergic transmission at other *Drosophila* NMJs (*DiAntonio, 2006*; *He and Dickman, 2025*), the expression of a previously characterized genome-edited GluRIIA-GFP fly strain in which GFP has been inserted near the carboxy-terminus of GluRIIA at its endogenous genomic location was evaluated in the male reproductive system. In the SV, GluRIIA-GFP (*Figure 8B and E*) exhibited punctate expression that paralleled, but was complementary to, that of the glutamatergic synaptic vesicle marker vGlut-40XV5 (*Figure 8A and D*). This distribution of GluRIIA-GFP is not unlike its distribution at the NMJ of third instar larva, optimally localized post-synaptically to bind glutamate released from pre-synaptic terminals (*Beckers et al., 2024*). Surprisingly, no GluRIIA-GFP was observed in the ED (*Figure 8H and K*) despite abundant expression of vGlut-40XV5 (*Figure 8G and J*). Expression of GluRIIA-GFP in the AGs (*Figure 8N and Q*) was similar to that in the SVs in that it closely paralleled vGlut-40XV5 expression (*Figure 8M and P*) but was complementary to it. Finding GluRIIA-GFP expression in the SVs and AGs, but not the ED, and the expression of GluRIIB in all three tissues based on the *GluRIIB-GAL4* expression pattern suggests the SVs and AGs use a mixture of glutamate receptors containing one or the other of the mutually exclusive GluRIIA or GluRIIB subunits in combination with the common GluRIIC, D, and E subunits, while all glutamate receptors in the ED solely utilize GluRIIB. As GluRII receptors containing GluRIIA generate higher currents than those that contain GluRIIB (*DiAntonio et al., 1999*), the lower GluRIIB currents are apparently sufficient for a functional ED.

## OA receptors exhibit distinct expression patterns in the muscles, neurons, and epithelial cells of the male reproductive system

OA receptors are modulatory G-protein coupled receptors (GPCRs) that mediate slower-acting but longer-lasting effects than ionotropic receptors. To visualize the expression of the OA receptors OAMB, OAα2R, and OAβ2R, CRISPR/Cas9 genome editing was used to create conditional alleles of each receptor fused at their carboxy termini to a multimerized epitope tag for enhanced detection sensitivity. These genome edits are at the endogenous genomic location of each gene and their expression is thus under the control of their complete endogenous regulatory regions. For visualizing expression in the male reproductive system, germline inversion or excision variants were generated that converted conditional expression to constitutive expression.

OAMB-10XV5 exhibited strong expression in the epithelial cells of the ED and AGs with limited expression in muscles (*Figure 8—figure supplement 1A*). Co-immunostaining with the synaptic vesicle marker Synapsin was conducted to visualize pre-synaptic terminals (*Figure 8—figure supplement 1B*).

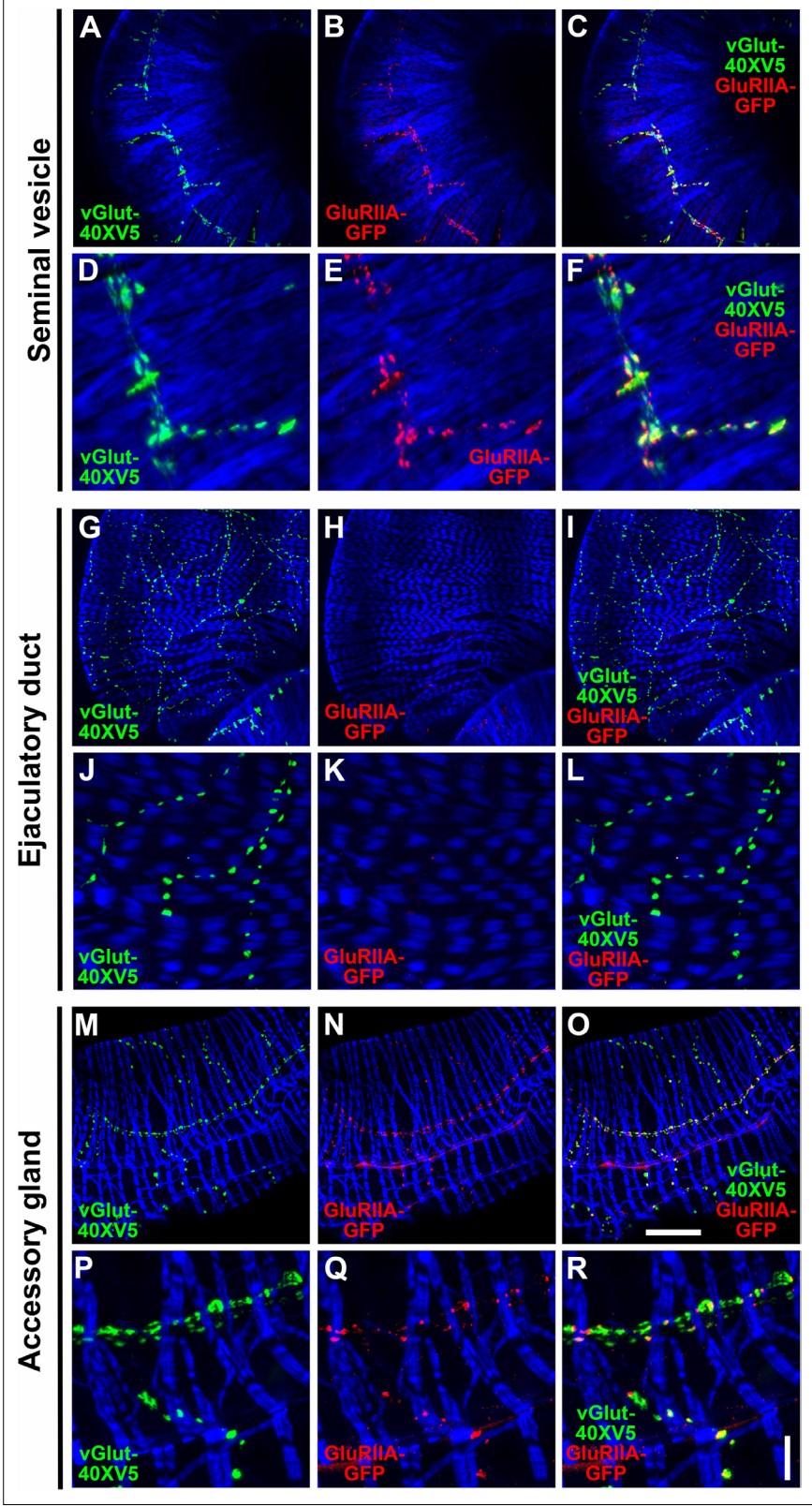

**Figure 8.** GluRIIA-GFP expression in the seminal vesicle (SV), ejaculatory duct (ED), and accessory gland (AG) of the *Drosophila* male reproductive system. (**A–F**) SV. (**A, D**) vGlut-40XV5; (**B, E**) GluRIIA-GFP; (**C, F**) vGlut-40XV5, GluRIIA-GFP overlay. (**G–L**) ED. (**G, J**) vGlut-40XV5; (**H, K**) GluRIIA-GFP; (**I, L**) vGlut-40XV5, GluRIIA-GFP overlay.

*Figure 8 continued on next page*

*Figure 8 continued*

(**M–R**) AG. (**M, P**) vGlut-40XV5; (**N, Q**) GluRIIA-GFP; (**O, R**) vGlut-40XV5, GluRIIA-GFP overlay. Scale bars: O-50 μm; R-10 μm.

The online version of this article includes the following figure supplement(s) for figure 8:

**Figure supplement 1.** OAMB expression in the *Drosophila* male reproductive system.

**Figure supplement 2.** OAMB expression in the *Drosophila* male reproductive system.

**Figure supplement 3.** OAα2R-20XV5 expression in the *Drosophila* male reproductive system.

**Figure supplement 4.** OAα2R-20XV5 expression in the *Drosophila* male reproductive system.

**Figure supplement 5.** OAβ2R-40XV5 expression in the *Drosophila* male reproductive system.

**Figure supplement 6.** OAβ2R-40XV5 expression in the *Drosophila* male reproductive system.

**Figure supplement 7.** OAβ2R-40XV5 expression in the *Drosophila* male reproductive system.

**Figure supplement 8.** 5-HT7-20XV5 expression in the *Drosophila* male reproductive system.

**Figure supplement 9.** 5-HT7-20XV5 expression in the *Drosophila* male reproductive system.

**Figure supplement 10.** 5-HT7-20XV5 expression in the *Drosophila* male reproductive system.

In the SVs, higher resolution images revealed low-level expression of OAMB-10XV5 in muscles as well as obvious expression in pre-synaptic neurons (*Figure 8—figure supplement 1D and G*) that largely overlapped with Syn (*Figure 8—figure supplement 1F and I*). OAMB-10XV5 was similarly distributed in the ED with low-level expression in muscle and clearly discernible expression in pre-synaptic neurons (*Figure 8—figure supplement 1J*) based on substantial overlap with Syn (*Figure 8—figure supplement 1L*).

Strong OAMB-10XV5 expression was also observed in the epithelial cells of the ED in a honeycomb pattern (*Figure 8—figure supplement 1M and P*) reminiscent of Dlg and presumably associated with the plasma membrane. OAMB-10XV5 expression in the epithelial cells of the AGs was similar to its expression in the epithelial cells of the ED with prominent signal in a honeycomb pattern (*Figure 8—figure supplement 2A and D*). OAMB-10XV5 also exhibited low-level expression in the muscles of the AG and expression in the pre-synaptic neurons (*Figure 8—figure supplement 2G*) by virtue of significant overlap with Syn (*Figure 8—figure supplement 2I*). Prominent OAMB-10XV5 expression was also observed in the ejaculatory bulb. Low-level expression of OAMB-10XV5 was detected in the muscles of the ejaculatory bulb, as was expression in the pre-synaptic neurons (*Figure 8—figure supplement 2J and M*) as evidenced by mostly overlapping expression with Syn (*Figure 8—figure supplement 2L and O*). In the non-muscular portion of the ejaculatory bulb, OAMB-10XV5 expression was readily discernible in the epithelial cells in a honeycomb pattern (*Figure 8—figure supplement 2J and P*).

OAα2R-20XV5 exhibited strong expression in the muscles of both the SVs and more distal regions of the proximal ED near where it connects with the ejaculatory bulb, as well as lower-level expression in the AGs (*Figure 8—figure supplement 3A*). Higher resolution images of OAα2R-20XV5 in the SV indicate expression is restricted to the muscle (*Figure 8—figure supplement 3G*) as there is no apparent overlap with Syn (*Figure 8—figure supplement 3I*). A cross-section of the SV muscle reveals OAα2R-20XV5 is localized to the membrane of the muscle (*Figure 8—figure supplement 3J*), as would be expected for a GPCR. In the AGs, OAα2R-20XV5 is expressed in the muscles (*Figure 8—figure supplement 3M and P*) as well as in the pre-synaptic neurons, as evidenced by co-localization with Syn (*Figure 8—figure supplement 3O and R*). OAα2R-20XV5 expression ends abruptly just after the SVs fuse into the sphincter that regulates the movement of sperm into the ED (*Figure 8—figure supplement 4A and D*). OAα2R is expressed in the pre-synaptic neurons innervating the sphincter with little to no expression in the sphincter muscles (*Figure 8—figure supplement 4C and F*). In the proximal region of the anterior ED, prominent OAα2R-20XV5 expression is observed throughout the innervating neurons with little to no OAα2R-20XV5 in the muscles (*Figure 8—figure supplement 4G, I, J, and L*). In contrast, in the distal region of the anterior ED, robust expression of OAα2R-20XV5 is detected in the muscles (*Figure 8—figure supplement 4M, P, and S*) but not in the pre-synaptic neurons (*Figure 8—figure supplement 4O, R, and U*). This complementary pattern of OAα2R-20XV5 in neurons but not muscles in the proximal ED and muscles but not neurons in the distal ED suggests these different regions of the ED are exhibiting a differential response to OA. As in the

SV, OAα2R-20XV5 localizes to the membrane of the muscles as evident in a cross-sectional image (*Figure 8—figure supplement 4S*). Interestingly, in this region, Syn expression is inside the muscles of the ED (*Figure 8—figure supplement 4T*), potentially suggesting the muscles are being stimulated from the inside, or alternatively, that the neurons are stimulating the underlying epithelial cells. As pre-synaptic neuronal expression of adrenoreceptors in mammals has been shown to be release-inhibiting (*Brown and Gillespie, 1957*; *Starke, 1972*), including in the urinary bladder (*Somogyi and de Groat, 1990*) and vas deferens (*O'Connor et al., 1999*; *Scheibner et al., 2001*), OAα2R expression in the pre-synaptic neurons of the internal male reproductive system may also function via inhibitory auto-reception.

OAβ2R-40XV5 exhibits strong expression in the epithelial cells of the ED, the muscles of the ejaculatory bulb, the muscles of the SVs near the point of fusion, the testes, with lower-level muscle expression in the SVs, ED, and AGs (*Figure 8—figure supplement 5A*). Higher resolution images of the SV reveal obvious expression in the muscle (*Figure 8—figure supplement 5D and G*) but not in the pre-synaptic neurons as there is no apparent overlap with Syn (*Figure 8—figure supplement 5F and I*). A cross-sectional image of the SV reveals OAβ2R-40XV5 is localized to the plasma membrane of muscle cells (*Figure 8—figure supplement 5J*). In higher resolution images of the ED, moderate levels of OAβ2R-40XV5 are present in the muscles (*Figure 8—figure supplement 5M and P*) but not in the innervating pre-synaptic neurons as there is no obvious overlap with Syn (*Figure 8—figure supplement 5O and R*). At the junction where the SVs fuse into a single duct, OAβ2R-40XV5 expression is strong prior to fusion but drops off abruptly at the point of full fusion (arrow, *Figure 8—figure supplement 6A*). The strongest levels of OAβ2R-40XV5 expression throughout the entire male reproductive system is in what appears to be the epithelial cells that line the ED (*Figure 8—figure supplement 6A*). A cross-section of the ED reveals the distribution of OAβ2R-40XV5 in epithelial cells is exclusively at the apical surface (*Figure 8—figure supplement 6D*). Also notable, there is another boundary of OAβ2R-40XV5 expression in the SVs somewhat prior to fusion (solid arrow, *Figure 8—figure supplement 6G*). Higher resolution images of OAβ2R-40XV5 expression at the point of SV fusion show the sharp dip in expression at the point of full fusion (arrow, *Figure 8—figure supplement 6J*). A cross-sectional image also reveals the drop in expression and membrane localization of OAβ2R-40XV5 in the muscle (*Figure 8—figure supplement 6M*). Similar to the proximal anterior ED, the more distal region of the anterior ED exhibits strong expression in the muscles, and especially the underlying epithelial cells (*Figure 8—figure supplement 7A and D*). In the AGs, OAβ2R-40XV5 expression is manifest in the muscles (*Figure 8—figure supplement 7G and J*) but does not overlap with the pre-synaptic neurons as evidenced by the lack of overlap with Syn (*Figure 8—figure supplement 7I and L*). Prominent OAβ2R-40XV5 expression was also observed in the ejaculatory bulb (*Figure 8—figure supplement 7M and P*). As β2 adrenoreceptors relax bladder muscles in mice (*Wuest et al., 2009*), OAβ2R may relax muscles in the male reproductive system.

Octopamine receptor expression in the internal organs of the male reproductive tract was predicted based on the dense OGN innervation of these tissues. Although OA receptor presence in these muscles aligns with their role in modulating contractile strength and rhythm, the organ-specific differences in receptor subtype abundance could only be established empirically. Epithelial expression implies that octopamine influences non-neuronal functions such as protein secretion or barrier permeability, but the precise cellular responses remain to be defined. Mapping OAMB, OAα2R, and OAβ2R to discrete muscle, neuronal, and epithelial compartments provides mechanistic information about how octopamine fine-tunes contraction, secretion, and presynaptic feedback across the male reproductive tract, offering an evolutionary framework for adrenergic control of fertility and pinpointing cell-specific targets for modulating reproductive output.

## 5-HT7 is widely expressed in the male reproductive system but most prominently in the sphincter region

Since the *5-HT7-GAL4* reporter delineates a sharp posterior boundary in the SV, endogenous 5-HT7 receptor expression was evaluated to pinpoint where 5-HT may directly modulate reproductive tract motility. To assess the expression pattern of the 5-HT7 receptor, a 20XV5 tag was added to the carboxy-terminus by CRISPR/Cas9 genome editing at its endogenous genomic location. 5-HT7-20XV5 localizes to the muscles of the SV just before and after the junction, the epithelial cells of the AGs, and the ejaculatory bulb (*Figure 8—figure supplement 8A*). Higher resolution images of the SV reveal

low-level expression in muscles (*Figure 8—figure supplement 8D and G*), as well as expression in the pre-synaptic neurons innervating the male reproductive system as evidenced by co-localization with Syn (*Figure 8—figure supplement 8F and I*). In the ED, 5-HT7-20XV5 is expressed in the muscles (*Figure 8—figure supplement 8J and M*), but unlike the SV, it is not expressed in the pre-synaptic neurons as it does not appear to co-localize with Syn (*Figure 8—figure supplement 8L and O*). Expression levels of 5-HT7-20XV5 increase in the SV near the point of fusion and are maximal from just before the junction all the way to the terminus in the ED (*Figure 8—figure supplement 9A*). Expression levels of 5-HT7-20XV5 appear to increase at a discrete boundary point on the SVs prior to fusion (arrow, *Figure 8—figure supplement 9D*). This boundary of increasing expression at this same region of the SV is highly reminiscent of OAβ2R-40XV5 (*Figure 8—figure supplement 6G*). A higher resolution image of the sphincter region of the SV post-fusion reveals high levels of 5-HT7-20XV5 expression (*Figure 8—figure supplement 9G*) but not in the innervating pre-synaptic neurons based on the absence of co-localization with Syn (*Figure 8—figure supplement 9I*). A cross-section of this same region reveals the 5-HT7-20XV5 protein localizes to muscle membranes (*Figure 8—figure supplement 9J*). A high-resolution image of the terminus of the sphincter confirms 5-HT7-20XV5 expression extends all the way to the end (*Figure 8—figure supplement 9M*). This observation of 5-HT7-20XV5 expression throughout the fused sphincter region of the SV contrasts with that of OAα2R-20XV5 (*Figure 8—figure supplement 4D*) and OAβ2R-40XV5 (*Figure 8—figure supplement 6A*), both of which end at the point of full SV fusion and suggests 5-HT plays a more important role than OA in sphincter regulation. 5-HT7-20XV5 expression is also elevated in the muscles at the terminus of the AGs (*Figure 8—figure supplement 9P*). In the AGs, 5-HT7-20XV5 is most prominently expressed in the underlying epithelial cells (*Figure 8—figure supplement 10A and D*). An AG cross-section reveals 5-HT7-20XV5 localizes exclusively to the apical surface of the epithelial cells (*Figure 8—figure supplement 10G*), highly reminiscent of OAβ2R-40XV5 localization to the apical surface of epithelial cells of the ED (*Figure 8—figure supplement 6D*). Nearer the surface of the AGs, low-level expression of 5-HT7-20XV5 is detected in the muscle (*Figure 8—figure supplement 10J*) and obvious expression is observed in the innervating pre-synaptic neurons based on overlapping expression with Syn (*Figure 8—figure supplement 10L*). 5-HT7-20XV5 expression was also observed in the ejaculatory bulb (*Figure 8—figure supplement 10M*). Interestingly, none of the OA or serotonin receptors exhibited preferential accumulation in the post-synaptic muscle opposite the pre-synaptic terminals, as observed for GluRIIA. This observation implies volume transmission may play a significant role in modulating muscle activity in the male reproductive system. Neurotransmitter signaling via volume transmission has previously been proposed for the *Drosophila* female reproductive system based on expression patterns of OA receptors (*Rohrbach et al., 2024a*).

## SGNs are essential for male fertility but OGNs are dispensable

Having established that both OGNs and SGNs innervate the male reproductive system, an obvious question follows: what are their roles in male fertility? It has previously been reported that OA-deficient flies are male fertile and female sterile (*Monastirioti et al., 1996*; *Cole et al., 2005*). However, in Tβh and Tdc2 mutant flies that fail to synthesize OA, glutamate signaling in the OGNs innervating the male reproductive system is still intact. To determine if simultaneously silencing the OGNs innervating the male reproductive system for both glutamate and OA signaling causes sterility, *Tdc2-GAL4-DBD*

**Table 1.** Male and female fertility phenotypes.

| Genotype | Male fertile | Female fertile |
|---|---|---|
| *yw; TRH-GAL4DBD/AbdB-AD, UAS-BONT-C* | No | No |
| *yw; Tdc2-GAL4DBD/AbdB-AD, UAS-BONT-C* | Yes | No |
| *yw; VT019028-GAL4-DBD/AbdB-AD, UAS-BONT-C* | No | Yes |
| *yw; TRH-AD; UAS-BONT-C/VT019028-GAL4-DBD* | No | Yes |
| *yw; AbdB-AD/UAS-BONT-C* | Yes | Yes |
| *yw; UAS-BONT-C* | Yes | Yes |
| *Canton-S* wild-type | Yes | Yes |

was combined with *AbdB-AD* and *UAS-BONT-C* to block all synaptic transmission in these neurons (BONT-C cleaves the SNARE protein syntaxin *Han et al., 2022*). This intersectional combination results in the silencing of only the Tdc2 neurons in the posterior VNC by virtue of the restricted expression of the AbdB-AD hemidriver to the posterior few segments of the VNC. The result was the same as eliminating OA signaling alone, males were fertile, and females were sterile (*Table 1*), thus establishing that neither OA nor glutamate signaling is required for fertility in the OGNs innervating the male reproductive system. The observed female sterility validates the silencing function of the UAS-BONT-C transgene.

To assess the requirement of the SGNs innervating the male reproductive system for fertility, flies containing TRH-DBD, AbdB-AD, and UAS-BONT-C were generated. Both males and females of this genotype were sterile (*Table 1*). The observation that females were also sterile, while no serotonergic innervation of the female reproductive system has been reported, suggests that male sterility is due to 5-HT neurons common to both males and females, and thus the male sterility cannot be unambiguously attributed to the SGNs innervating the male reproductive system.

In an attempt to identify a split-GAL4 combination that more specifically targets SGNs innervating the male reproductive system, several dozen hemi-drivers that express in neurons whose cell bodies are located in the most posterior region of the VNC were screened for male sterility in combination with UAS-BONT-C and either *TRH-AD* or *TRH-GAL4-DBD*, as appropriate. One line, *VT019028-GAL4-DBD* was identified that resulted in male sterility and female fertility, suggesting silencing of male-specific sexually dimorphic neurons, and not neurons common to both males and females, were responsible for the male sterility. The *TRH-AD/VT019028-GAL4-DBD* driver in combination with a *UAS-His2A-GFP* nuclear reporter results in expression in sparse neurons in the brain and VNC, as well as around two dozen neurons at the posterior tip of the VNC (*Figure 9A*). Female flies of the same genotype exhibited a similar sparse pattern of expression in the brain and VNC with the exception that there were noticeably fewer His2A-GFP-expressing neurons at the tip of the VNC (*Figure 9B*). The neurons most likely responsible for male sterility are those unique to males. The *VT019028-GAL4-DBD* driver was also assessed for fertility in combination with *AbdB-AD* and *UAS-BONT-C* with the same result of male sterility and female fertility (*Table 1*). The expression patterns of *VT019028-GAL4-DBD* with *AbdB-AD* in combination with *UAS-His2A-GFP* reveal restricted expression to the tip of the ventral nerve cord with noticeably more neurons in the male (*Figure 9A, C and D*). These results with *AbdB-AD* narrow the neurons responsible for male sterility to the tip of the VNC. The expression patterns of 5-HT and Tdc2 in these immunostains are also shown (*Figure 9E-H and I-L*, respectively).

To determine if any of the *VT019028-GAL4-DBD* neurons intersecting with *TRH-AD* or *AbdB-AD* are serotonergic or octopaminergic, higher resolution images of the tip of the VNC were examined. In the full confocal stack image, three His2A-GFP neurons are clearly visible that overlap with 5-HT (*Figure 9—figure supplement 1D*). However, it has previously been shown that there are both dorsal (~10 neurons) and ventral (~9 neurons) clusters of serotonergic neurons innervating the male reproductive system (*Billeter et al., 2006b*) and some neurons may thus be obscured from a dorsal perspective. A partial stack containing the dorsal cluster reveals the same three neurons (*Figure 9—figure supplement 1I*). However, a ventral view of these neurons shows a fourth neuron overlapping with 5-HT (*Figure 9—figure supplement 1K*). A partial stack of the ventral cluster reveals three His2A-GFP neurons overlapping with 5-HT (*Figure 9—figure supplement 1O*). There also appears to be one or two Tdc2 neurons overlapping with His2A-GFP. However, examination of three-dimensional images revealed that they are only in close proximity but not overlapping.

Similar results were obtained with *VT019028-GAL4-DBD* neurons intersecting with *AbdB-AD*. Four His2A-GFP neurons overlap with 5-HT in a full confocal stack view (*Figure 9—figure supplement 2D*). These four neurons are visible in the dorsal serotonergic cluster (*Figure 9—figure supplement 2I*), while three additional His2A-GFP neurons overlapping with 5-HT are apparent in the ventral serotonergic neuron cluster (*Figure 9—figure supplement 2N*). One His2A-GFP neuron appears to overlap with Tdc2 (*Figure 9—figure supplement 2E*), but this apparent overlap is not reproduced in the dorsal (*Figure 9—figure supplement 2J*) or ventral (*Figure 9—figure supplement 2O*) partial stacks. The innervation of *VT019028-GAL4-DBD* neurons with *TRH-AD* and *AbdB-AD* visualized with the *UAS-CD8-mCherry* reporter is similarly broad in the SVs, ED, and AGs (*Figure 9—figure supplement 3A and B*, respectively). Taken together, these results strongly suggest synaptic transmission in the SGNs included in the *VT019028-GAL4-DBD* intersections with *TRH-AD* and *AbdB-AD* that innervate

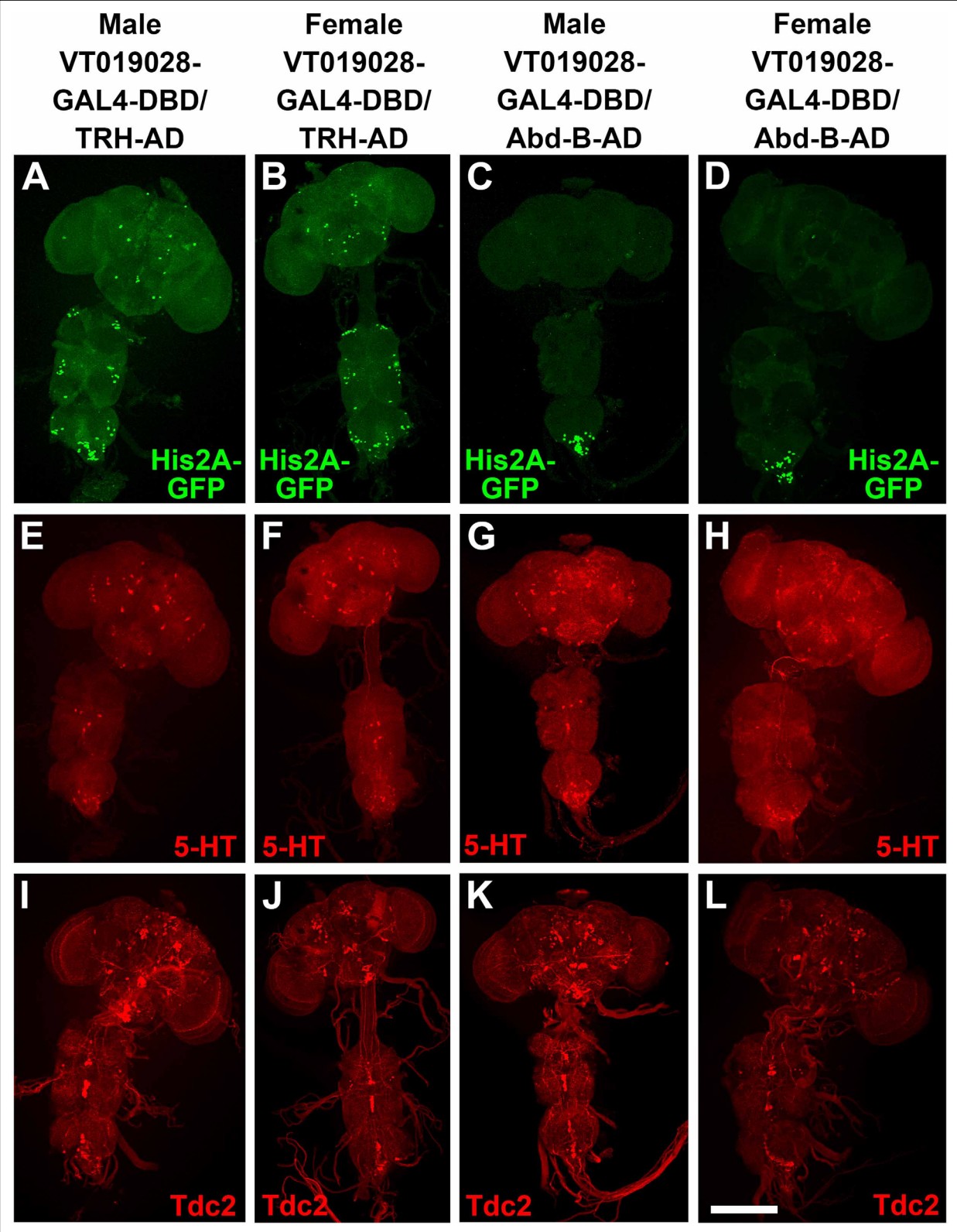

**Figure 9.** Low-resolution images of nuclear expression of *VT019028-GAL4-DBD* in combination with *TRH-AD* and *AbdB-AD* in *Drosophila* male and female adult nervous system relative to 5-HT and Tdc2. (**A–D**) nuclear expression using HIS2A-GFP marker. (**A**) male *VT019028-GAL4-DBD/TRH-AD*; (**B**) female *VT019028-GAL4-DBD/TRH-AD*; (**C**) male *VT019028-GAL4-DBD/AbdB-AD*; (**B**) female *VT019028-GAL4-DBD/AbdB-AD*. (**E–H**) 5-HT expression. (**E**) male *VT019028-GAL4-DBD/*TRH-AD; (**F**) female *VT019028-GAL4-DBD/TRH-AD*; (**G**) male *VT019028-GAL4-DBD/AbdB-AD*; (**H**) female *VT019028-*

*Figure 9 continued on next page*

Figure 9 continued

*GAL4-DBD/AbdB-AD*. Tdc2 expression. (**I**) male *VT019028-GAL4-DBD/TRH-AD*; (**J**) female *VT019028-GAL4-DBD/TRH-AD*; (**K**) male *VT019028-GAL4-DBD/AbdB-AD*; (**L**) female *VT019028-GAL4-DBD/AbdB-AD*. Scale bar: 200 μm.

The online version of this article includes the following figure supplement(s) for figure 9:

**Figure supplement 1.** High-resolution images of nuclear expression of *VT019028-GAL4-DBD* in combination with *TRH-AD* in posterior ventral nerve cord of *Drosophila* male adult nervous system relative to 5-HT and Tdc2.

**Figure supplement 2.** High-resolution images of nuclear expression of *VT019028-GAL4-DBD* in combination with *TRH-AD* in posterior ventral nerve cord of *Drosophila* male adult nervous system relative to 5-HT and Tdc2.

**Figure supplement 3.** High-resolution images of nuclear expression of *VT019028-GAL4-DBD* in combination with *AbdB-AD* in posterior ventral nerve cord of *Drosophila* male adult nervous system relative to 5-HT and Tdc2.

the male reproductive organs are required for male fertility, although a role for the small number of other interneurons at the tip of the VNC in common between both combinations cannot be excluded.

To rule out the possibility that the male sterility observed in the *VT019028-GAL4-DBD* intersections with *UAS-BONT-C* and either *TRH-AD* and *AbdB-AD* is due to failure to mate, mating assays were performed where single males and single females of the different genotypes were placed together in a well of a 12-well plate and videotaped. The mating success rates of *VT019028-GAL4-DBD/TRH-AD* and *VT019028-GAL4-DBD/AbdB-AD* were 81.25% (n=16) and 57.9% (n=28), respectively. These results demonstrate the observed male sterility is not due to failure to copulate.

In these mating assays, mating duration was also assessed along with several control genotypes. Mean mating durations for *VT019028-GAL4-DBD/TRH-AD* and *VT019028-GAL4-DBD/AbdB-AD* were 28.74 and 24.75 min, respectively (***Figure 10A***). This was not significantly different from *Canton-S* wild-type controls (26.91), and other controls missing one or more of the components necessary for silencing *AbdB-AD*, *UAS-BONT-C* (27.07), and *UAS-BONT-C* alone (28.75) (***Figure 10B and C***). These results are in line with previously reported mating durations for *Drosophila* (***Crickmore and Vosshall, 2013***). Interestingly, silencing all 5-HT neurons (*TRH-GAL4-DBD Ç AbdB-AD*) or all OA neurons (*Tdc2-GAL4-DBD Ç AbdB-AD*) innervating the male reproductive system resulted in statistically significant reductions in mating duration, 19.11 (***Figure 10D***) and 21.8 (***Figure 10E***), respectively.

To assess the distribution of sperm in the reproductive system during mating by *VT019028-GAL4-DBD/TRH-AD>UAS-BONT-C* sterile males, a Protamine-GFP (Prot-GFP) reporter that fluorescently labels sperm nuclei was incorporated into the genotype. For this experiment, control and experimental males were placed in petri dishes with virgin females, separated 10 min after the initiation of mating, immediately dissected, and the isolated male reproductive systems video recorded using a stereofluorescent microscope. In the control males, Prot-GFP sperm were readily visible in the ED (***Figure 10—figure supplement 1A***, ***Video 1***). In contrast, in the experimental males, no Prot-GFP sperm were observed (***Figure 10—figure supplement 1***, ***Video 2***). ProtB-GFP expression in the reproductive systems of wild-type Canton-S females mated to these same male genotypes was also assessed. In this experiment, matings were allowed to go to completion and the female reproductive systems were dissected immediately thereafter. An abundance of Prot-GFP sperm were observed in females mated to the control males (***Figure 10—figure supplement 1C***), while no Prot-GFP sperm were observed in females mated to *VT019028-GAL4-DBD/TRH-AD>UAS-BONT-C* males (***Figure 10—figure supplement 1D***). These results demonstrate that male sterility is due to a failure of the sphincter that regulates the flow of sperm between the SV and ED to open, and thus the defect is with the emission phase of ejaculation.

## Glutamate signaling in OGNs and SGNs is dispensable for male fertility

In the *VT019028-GAL4-DBD/TRH-AD>UAS-BONT-C* sterile males, 5-HT and glutamate signaling are both abolished due to the BONT-C blockade of synaptic transmission. To determine if glutamate signaling alone is required in these neurons, a conditional allele of vGlut in which the vGlut coding sequence is excised by recombination (***Sherer et al., 2020***) was incorporated into this genotype, such that specifically glutamate signaling was eliminated while serotonergic signaling remained intact. These males were determined to be fertile. To ascertain whether there were changes in the level of fertility, fertility assays were conducted on 5 day post-eclosion males. No changes in fertility as compared to controls were observed (***Figure 10—figure supplement 2A***). Immunostaining the AGs of control

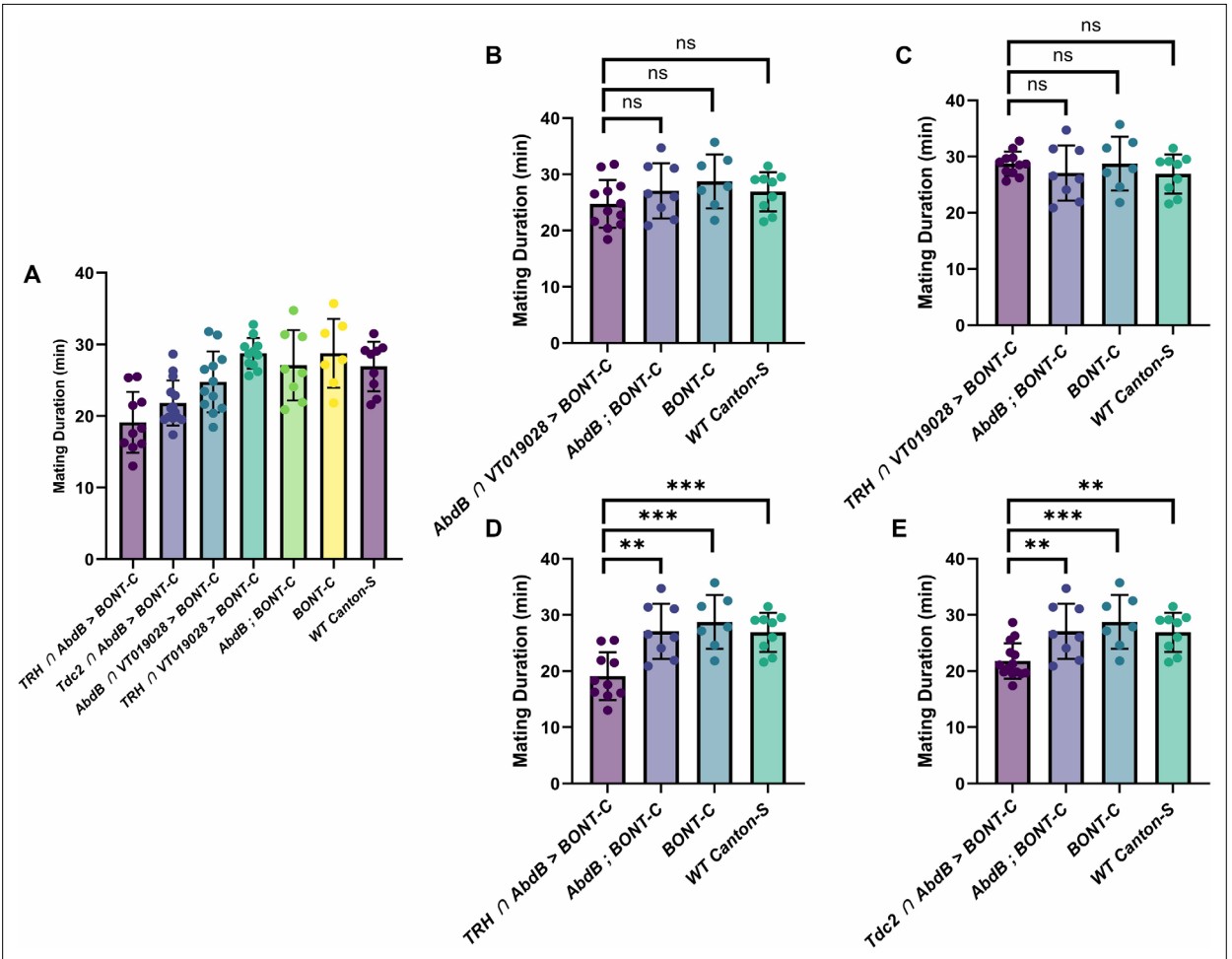

**Figure 10.** Comparison of mating duration between genotypes. Bar plots show the mean ± SD for each group with individual data points overlaid on each bar. Statistical comparisons were made using unpaired two-tailed t tests where appropriate. Asterisks above indicate significance levels: $p<0.001$ (**), $p<0.0001$ (***), ns = not significant. (**A**) Columns from left to right: TRH ∩ AbdB >BONT-C: n=10, mean = 19.11, SD = 4.251. Tdc2 ∩ AbdB >BONT-C: n=14, mean = 21.80, SD = 3.155. AbdB ∩ VT019028>BONT-C: n=12, mean = 24.75, SD = 4.247. TRH ∩ VT019028>BONT-C: n=11, mean = 28.74, SD = 2.155. AbdB; BONT-C: n=8, mean = 27.07, SD = 4.907. BONT-C: n=7, mean = 28.75, SD = 4.809. WT Canton-S: n=9, mean = 26.91, SD = 3.473.

The online version of this article includes the following source data and figure supplement(s) for figure 10:

**Source data 1.** Mating duration data of control and experimental genotypes.

**Figure supplement 1.** Prot-GFP fluorescent marker in male and female reproductive systems in control and *TRH-AD/VT019028-GAL4-DBD* neuron-silenced flies.

**Figure supplement 2.** Comparison of male *Drosophila* fecundity between genotypes.

**Figure supplement 2—source data 1.** Five day old male fertility data of control and experimental genotypes.

**Figure supplement 2—source data 2.** 30 day old male fertility data of control and experimental genotypes.

**Figure supplement 3.** vGlut expression in accessory glands (AGs) of control and vGlut conditional mutant.

and experimental flies confirmed the effectiveness of the conditional vGlut allele by the presence of vGlut in control flies and its absence in experimental flies (*Figure 10—figure supplement 3A and D*, respectively). As glutamate has been shown to promote neuron survival as neurons age (*Shen et al., 2018*; *Buck et al., 2023*), to determine if fertility might be altered in older males, the same 5 day-old males used in the previous fertility experiment were held in isolation from females until they were 30 days post-eclosion and the experiment was repeated. While there was not unexpectedly a modest decrease in fertility with age, no differences in fertility were apparent between the control and experimental genotypes (*Figure 10—figure supplement 2B*). These results establish that glutamatergic

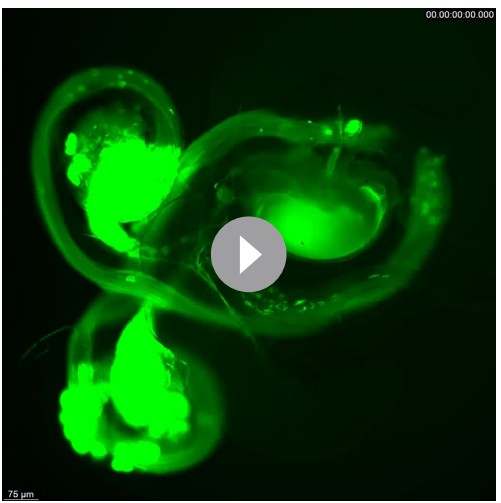

**Video 1.** Male reproductive system of a ProtB-GFP control. Sperm with green fluorescent nuclei are visible in the ejaculatory duct of a male subjected to a mid-mating interruption.

https://elifesciences.org/articles/108225/figures#video1

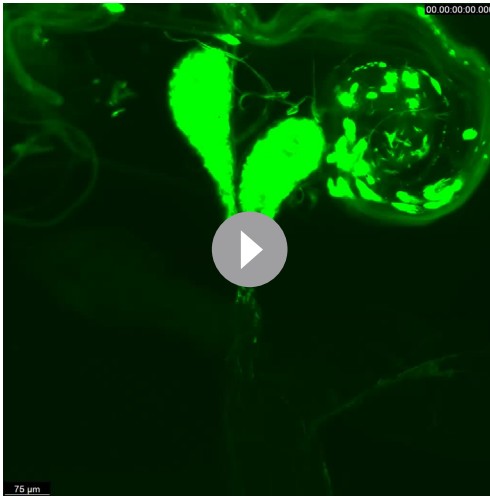

**Video 2.** Male reproductive system of a ProtB-GFP experimental male in which a subset of the *TRH-AD/VT019028-GAL4-DBD* subset of serotonergic neurons innervating the male reproductive system have been silenced. Sperm with green fluorescent nuclei are restricted to the seminal vesicle and never enter the ejaculatory duct from a male subjected to a mid-mating interruption.

https://elifesciences.org/articles/108225/figures#video2

signaling is not required in *VT019028-GAL4-DBD/TRH-AD* neurons for fertility. To determine if more extensive elimination of glutamate signaling in the OGNs and SGNs would alter fertility, glutamate was specifically eliminated in both classes of neurons. No reductions in fertility were observed in 5 day post-eclosion males (*Figure 10—figure supplement 2A*). These results indicate glutamate is not required in either OGNs or SGNs for fertility.

## The sphincter region of the SV initiates ED peristaltic waves

Neuron-independent peristaltic waves are known to occur spontaneously in the male ED of *Drosophila* that persist even when all nerves innervating the male reproductive system have been severed (*Norville et al., 2010*). To gain a better understanding of the muscle activity occurring during these waves in both the ED and throughout the rest of the male reproductive system, a fly strain was generated that expressed the Ca++ sensor GCAMP8m specifically in muscles under control of the myosin heavy chain promoter (*MHC-GCAMP8m*). This fly strain allows visualization of the changes in Ca++ levels that occur in the muscles during these peristaltic waves. As expected, waves of fluorescence are visualizable coincident with peristaltic waves of muscle contractions in the ED (*Video 3*). These peristaltic waves originate not in the ED but in the sphincter region of the SV just prior to and after the point of SV fusion (*Video 4*), although the region of the SV prior to the fusion point that is involved in initiation varies (*Video 5*). Muscle contractions in other regions of the male reproductive system

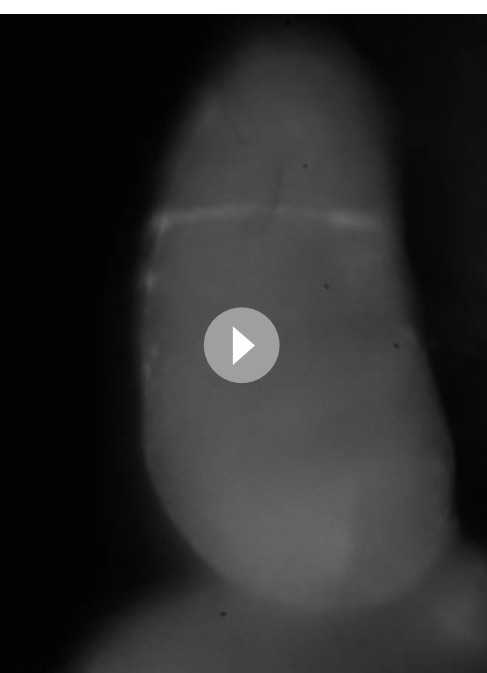

**Video 3.** GCAMP8m Ca++ imaging of an ejaculatory duct experiencing spontaneous peristaltic waves of muscle contractions.

https://elifesciences.org/articles/108225/figures#video3

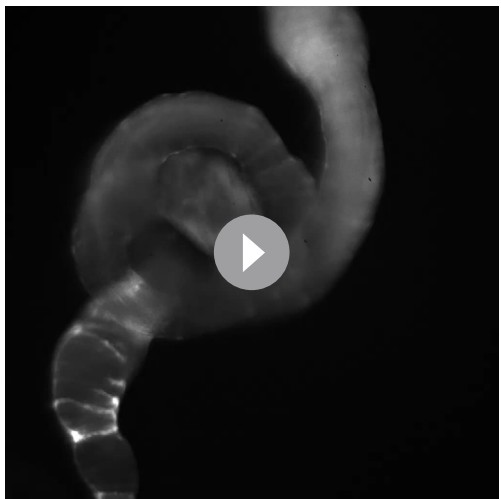

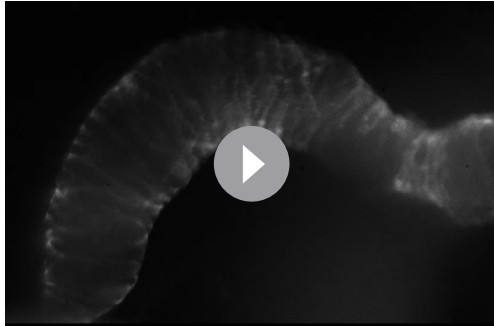

**Video 5.** GCAMP8m Ca++ imaging of the seminal vesicle (SV)/ejaculatory duct (ED) junction showing the subtle variation in the initiation site of spontaneous peristaltic waves as compared to *Video 4*.

https://elifesciences.org/articles/108225/figures#video5

**Video 4.** GCAMP8m Ca++ imaging of the seminal vesicle (SV)/ejaculatory duct (ED) junction showing the initiation site of the spontaneous peristaltic waves.

https://elifesciences.org/articles/108225/figures#video4

are also apparent. These contractions are weak and seemingly random in the SV (*Video 6*), AG (*Video 7*), and testes (*Video 8*), but occasionally in the AG, strong coordinated contractions are observed (*Video 9*). Together, these imaging data reveal that the seminal-vesicle sphincter acts as an autonomous pacemaker, launching neuron-independent peristaltic waves that drive ejaculatory-duct motility.

## Discussion

The male reproductive tract of flies and mammals have obvious morphological differences, yet they share many essential anatomical features and a common organizational logic. During copulation, *Drosophila* males deliver seminal fluid and ~4000–6000 sperm to females during a tightly choreographed ~20 min window (**Kaufman and Demerec, 1942**; **Gilbert, 1981**). This involves the emission of sperm and seminal fluid from their sites of origin into the ED where they mix and are subsequently expelled into the female reproductive tract. Despite the critical importance of ejaculation, its neuronal mechanisms have remained poorly defined, hindering broader insights into the neural logic of male fertility. Here, we close this gap by demonstrating that two distinct yet intermingled classes of amine/glutamate neurons, OGNs and SGNs, coordinate the ejaculatory sequence that ensures successful reproduction. Their transmitter pairs confer the capacity to co-release both fast excitatory and slower-acting aminergic modulatory neurotransmitters. Differential expression of ionotropic glutamate receptors and aminergic metabotropic neurotransmitter receptors combines in post-synaptic muscle across the male reproductive system to confer an additional level of complexity to coordinately regulate fluid secretion, organ contractility, and directional sperm movement, to the end of optimizing male fertility.

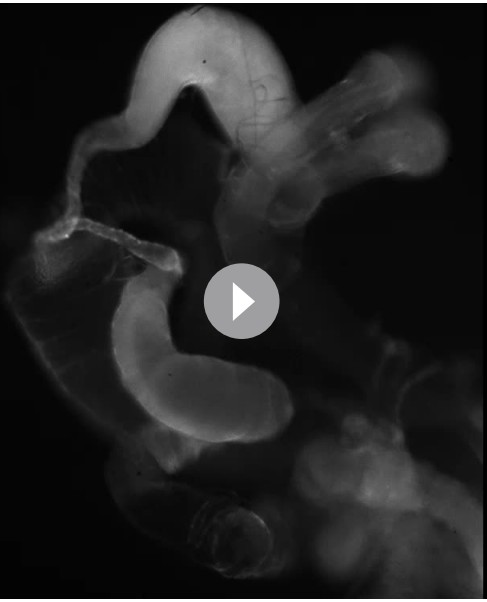

**Video 6.** GCAMP8m Ca++ imaging of the seminal vesicle (SV) showing spontaneous uncoordinated muscle activity.

https://elifesciences.org/articles/108225/figures#video6

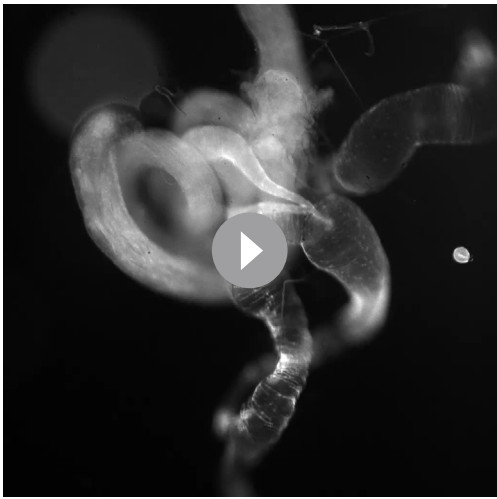

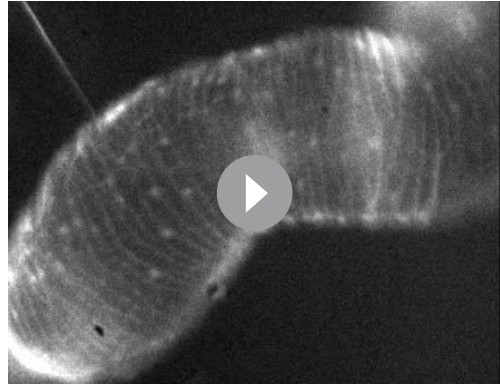

**Video 8.** GCAMP8m Ca++ imaging of a testes showing spontaneous uncoordinated muscle activity.
https://elifesciences.org/articles/108225/figures#video8

**Video 7.** GCAMP8m Ca++ imaging of an accessory gland (AG) showing spontaneous uncoordinated muscle activity.
https://elifesciences.org/articles/108225/figures#video7

This system can serve as a genetically tractable model to gain general insights into how multi-transmitter neurons coordinate and modulate interrelated action sequences critical for complex organ function. Lessons learned from this system may have implications for understanding how males of *Drosophila* and other species adapt their reproductive strategies to a dynamic environment. Additionally, understanding the mechanisms by which this system operates has the potential to identify neuromodulation strategies for male infertility and prostate cancer, as multi-transmitter sympathetic and parasympathetic neurons innervating human male reproductive organs have been implicated in these conditions (*Burnstock, 2009*; *Magnon et al., 2013*; *Burnstock, 2014*; *Drobnis and Nangia, 2017*; *Dwivedi et al., 2021*).

## Mechanisms of multi-transmitter neuron-mediated transfer of sperm and seminal fluid

Movement of sperm and seminal fluid through the male reproductive tract requires an alternating series of coordinated muscle contractions and relaxations. Based on the observed co-packaging of vMAT and vGlut in the majority of synaptic vesicles in the ED of both the OGNs and SGNs, combined with the observed non-uniform distribution of neurotransmitter receptors for OA, 5-HT, and glutamate in neurons and muscles, potential pre- and post-synaptic mechanisms for regulating the activity of the muscles of the internal male reproductive system are possible.

Pre-synaptic expression of the α-type OA receptors OAMB and OAα2R may facilitate alternating bursts of action potential-dependent neurotransmitter release by diminishing pre-synaptic neurotransmitter release via an auto-receptive mechanism. As α-type adrenergic and noradrenergic receptors have been consistently shown to inhibit neurotransmitter release in multiple rodent species in both the peripheral and central nervous system, including the vas deferens, heart, and brain when expressed in pre-synaptic neurons (*Starke, 1972*; *Trendelenburg et al., 1994*; *Miyazaki et al., 1998*; *Scheibner et al., 2001*; *Trendelenburg et al., 2003*), the presence of OAMB and OAα2R in the pre-synaptic terminals of neurons innervating the

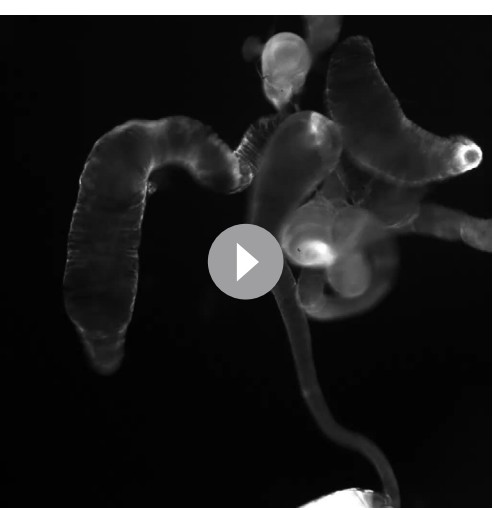

**Video 9.** GCAMP8m Ca++ imaging of the accessory gland (AG) showing strong coordinated muscle activity.
https://elifesciences.org/articles/108225/figures#video9

internal male reproductive organs may have the effect of suppressing neurotransmitter output from pre-synaptic terminals and shortening the duration of muscle contractions. Thus, pre-synaptic expression of OAMB and OAα2R may have the effect of increasing the cycling frequency of muscle contractions and relaxations.

As glutamate is acting via post-synaptic GluRII-type receptors, its effect will almost certainly be excitatory with the result of eliciting fast muscle contractions. In contrast, the functional consequences of OA and 5-HT are not so clear-cut. Either of these amines could enhance muscle contractions or facilitate muscle relaxation, depending on the neurotransmitter receptor(s) receiving the signal. In muscles of the mammalian prostate and lower urinary tract, α-type adrenergic and noradrenergic receptors have been demonstrated to augment muscle contractions while β-type receptors have been found to facilitate muscle relaxation (*Honda et al., 1985*; *Michel and Vrydag, 2006*; *Wuest et al., 2009*). In contrast, OAMB and OAβ2R have been shown to enhance muscle relaxation and contraction, respectively, in the *Drosophila* female reproductive system (*Deshpande et al., 2022*).

5-HT7 receptors are known to promote smooth muscle relaxation in the urinary tract, gut, oviduct, and blood vessels of mammals (*Carter et al., 1995*; *Leung et al., 1996*; *Prins et al., 1999*; *Inoue et al., 2003*; *Recio et al., 2009*; *Chang Chien et al., 2015*; *Matsumoto-Miyai et al., 2015*). Although muscle contraction receives preferential focus, muscle relaxation is also critically important for the execution of motor programs (*Goulding et al., 2014*; *Wong and Lange, 2014*; *Gowda et al., 2018*; *Hiramoto et al., 2021*) and 5-HT7 most likely plays a role in mediating muscle relaxation in the internal male reproductive system. An additional confounding factor is that OAα2R has been shown to respond not just to OA but also to 5-HT (*Qi et al., 2017*). The observations of the presence of both muscle contraction enhancing and muscle relaxation promoting, OA and 5-HT receptors in the same muscles of the internal male reproductive organs indicate the mechanisms of neuronal regulation of their activity are complex and may even involve synergistic effects that can only be established empirically.

## A potential timing mechanism mediated by multi-transmitter neurons

Pre-synaptic release of glutamate acting on post-synaptic ionotropic GluRII-type glutamate receptors will cause near immediate, short-term, contraction of muscles. At least in the ED, our results show that the majority of synaptic vesicles co-package either OA or 5-HT along with glutamate, and thus simultaneous co-release is being used as a mechanism of neurotransmitter signaling. The amines OA or 5-HT acting on post-synaptic metabotropic OA and 5-HT receptors will have the effect of either prolonging the length of, or augmenting the relaxation of, muscles, dependent on the specific aminergic neurotransmitter receptors involved. These latter effects will be delayed relative to the glutamate-induced fast muscle contractions due to the discrepancy in the time course of action between ionotropic and metabotropic receptors. Thus, co-releasing a fast-acting excitatory neurotransmitter acting through an ionotropic receptor simultaneous with a modulatory aminergic neurotransmitter acting through slower-acting GPCRs in the same multi-transmitter neuron may create a sequential timing mechanism whereby fast glutamate-mediated muscle contractions always precede either amine-mediated prolonging of muscle contraction or facilitation of muscle relaxation. A single neuron co-releasing one fast and one slow neurotransmitter may thus have the advantage over two distinct neurons separately releasing these transmitters in that the former can more effectively synchronize, and at the same time, stagger events that require precise timing.

## Potential for adaptation to environment

The complexity of neuronal control of male reproductive function revealed in this investigation exposes a variety of possibilities for adaptive regulation to dynamic environmental conditions. Polygynous *Drosophila* males are well-established to alter the quantity and quality of their ejaculate based on environmental circumstances, especially the level of competition with other males (*Bretman et al., 2009*; *Moatt et al., 2014*), but also due to nutrient availability (*Macartney et al., 2021*) and the presence of pathogens (*Liao et al., 2024*). The male reproductive system may adapt to environmental conditions by tuning octopamine, serotonin, and glutamate signaling, adjusting the levels of their biosynthetic enzymes, the vesicular glutamate and monoamine transporters vGlut and vMAT, or relevant receptors to alter the amount of sperm and seminal fluid released during copulation. Given that glutamate-deficient SGNs and OGNs, as well as BONT-C-silenced OGNs, show no readily detectable

male fertility defects under benign laboratory conditions, the primary role of glutamate and octopamine may be to optimize male reproductive success under fluctuating or adverse environmental conditions.

An additional potential mechanism for altering OA and/or 5-HT levels is extra-synaptic volume transmission, whereby the level of OA or 5-HT secreted into the hemolymph in locations anatomically distant from the male reproductive system. Thus, global levels of OA and 5-HT in the hemolymph could be adjusted based on environmental conditions and this signal could then impact the male reproductive system to optimize ejaculate quality and quantity. Finally, the presence of OAMB and 5-HT7 on epithelial cells suggests the potential for the regulation of the quality of seminal fluid, as the expression of these receptors implies, they could differentially alter gene expression in the epithelial cells in response to changing levels of OA and 5-HT. The results of this study create a foundation for mechanistic investigations into the molecular basis by which male reproductive resources are adaptively modulated.

## Potential for signaling via neuropeptides

The broad abundant expression of the LDCV reporter IA2 in the neurons innervating the internal organs of the male reproductive system suggests a significant role for neuropeptide signaling since neuropeptides are known to be packaged into LDCVs (*Pelletier and Leclerc, 1979*; *Kreiner et al., 1986*). Bolstering the prospect of this possibility, numerous neuropeptides across evolutionary distant insect species have been shown to alter the frequency and/or amplitude of the spontaneous peristaltic waves in the ED of male insect reproductive systems (*Marciniak et al., 2017*; *Čižmár et al., 2019*; *Lange et al., 2023*). Multiple neuropeptides have also been implicated in the function of the internal male reproductive organs of mammals, including humans (*Segawa et al., 1978*; *Allen et al., 1982*; *Tainio, 1995*). Given these considerations, it thus seems likely that neuropeptides outside the scope of this investigation also play a role in regulating *Drosophila* male reproduction.

## The SV/ED junction is a control center

Immediately proximal to the ED, SVs fuse into a single duct that forms the sphincter regulating sperm flow from the SV to the ED during mating. The differential expression patterns of OAα2R, OAβ2R, and 5-HT7 in this region of the SV is intriguing. The extent of OAα2R and OAβ2R expression ends very shortly after the merge, at precisely the point where full SV fusion occurs and there is no longer a partition separating the ducts (*Bairati, 1968*). Only 5-HT7 expression extends into the fully fused SV that constitutes the sphincter separating the SV from the ED. This expression pattern implies a prominent role for 5-HT7 in sphincter regulation. Previous reports of 5-HT7 mediating relaxation of smooth muscle in the mammalian gut (*Carter et al., 1995*; *Prins et al., 1999*), urinary bladder neck (*Recio et al., 2009*), cardiovascular system (*Leung et al., 1996*; *Chang Chien et al., 2015*), lymphatic system (*Chan and von Der Weid, 2003*), and oviduct (*Inoue et al., 2003*) raise the possibility it may function to relax the sphincter muscles during copulation to permit sperm flow, consistent with our observations that silencing a subset of serotonergic neurons results in male fertility due to failure of sperm emission into the ED (*Figure 10—figure supplement 1*). Additionally, the expression of OAβ2R and the likely relaxation-promoting 5-HT7 receptors increases sharply in the SV just prior to the point of fusion. Perhaps not coincidentally, this limit of expression corresponds with the distal-most extent of sperm localization in the SV where relaxation of the SV muscles must also occur for sperm to move along the duct to the ED. It is also notable, as revealed in the Ca++ imaging experiments, that the sphincter region is the site of initiation of the peristaltic waves of muscle contractions that are promulgated through the ED. The SV/ED junction is thus a critical control center for both regulation of sperm flow into the ED during mating and initiating neuron-independent peristaltic waves in the ED.

## Molecular continuity of neurotransmitter signaling between males and females

The expression of 5-HT7 and OAβ2R on the apical surface of the epithelial cells lining the AG and ED, respectively, is surprising because it implies that levels of 5-HT and OA are fluctuating in the lumen of these organs. The origin of these neurotransmitters is not obvious but if OA and 5-HT are present in the lumen of the AG and ED, another intriguing possibility is that these neurotransmitters are transferred to the female at mating for signaling in order to modulate female behavior and/or reproductive

activity. A recent study detailed the molecular continuity of proteins in the seminal fluid of males functionally interacting with proteins in the female reproductive tract (*McCullough et al., 2022*). The potential presence of OA and 5-HT in the lumen of the AG and ED raises the possibility that this molecular continuity could extend to neurotransmitters in male seminal fluid signaling to neurotransmitter receptors in the female reproductive tract to convey information such as health, internal state, or environmental conditions to regulate female reproductive output. A detailed examination of the expression patterns of OA receptors in the female reproductive tract indicates this is indeed a possibility, at least for OA signaling (*Rohrbach et al., 2024b*).

## Materials and methods
### Fly strains

Stocks from the Bloomington *Drosophila* Stock Center (NIH P40OD018537) were used in this study. *Tdc2-GAL4* (*Cole et al., 2005*) RRID:BDSC_9313, *GluRIIA-GAL4* RRID:BDSC_84637, *GluRIID-GAL4* RRID:BDSC_84638, and *TRH-GAL4* RRID:BDSC_84694 (*Deng et al., 2019*)
*vGlut-GAL4* RRID:BDSC_60312, *vGlut-GAL4-DBD* (*Diao et al., 2015*)
*Tdc2-AD* (*Dionne et al., 2018*) RRID:BDSC_601902
*TRH-GAL4-DBD* (*Aso et al., 2014*) RRID:BDSC_70371
*TRH-AD* (*Aso et al., 2014*) RRID:BDSC_70975
*GluRIIB-GAL4* RRID:BDSC_60333 and *GluRIIE-GAL4* RRID:BDSC_60332 (*Diao et al., 2015*)
*Oct-TyrR-GAL4* RRID:BDSC_77735 and *GluRIIC-GAL4* RRID:BDSC_83268 (*Lee et al., 2018*)
*GluRIIA-GFP* RRID:BDSC_99517 (*Beckers et al., 2024*)
*5-HT1A-GAL4* RRID:BDSC_84588, *5-HT1B-GAL4* RRID:BDSC_84589, *5-HT2A-GAL4* RRID:BDSC_84590, *5-HT2B-GAL4* RRID:BDSC_84591, and *5-HT7-GAL4* RRID:BDSC_84592 (*Deng et al., 2019*)
*TbH-GFP* RRID:BDSC_80113 (*Li Kroeger et al., 2018*)
*fru-GAL4* RRID:BDSC_66696
*UAS-CD8-mCherry* RRID:BDSC_27392
*ProtB-GFP* (*Manier et al., 2010*) RRID:BDSC_58406
*AbdB-AD* (*Diao et al., 2024*) *vGlut-AD* (*Gao et al., 2008*)
*IA2-GFP* (*Yu et al., 2025*)
*UAS-BONT-C* (*Han et al., 2022*)
*dsx-GAL4-DBD* (*Shirangi et al., 2016*)
*vGlut-40XV5* and *vGlut-40XMYC* (*Stowers, 2025*)
*B2RT-7XMYC-vAChT* (*Tison et al., 2020*)
*RSRT-9XV5-vGAT* (*Certel et al., 2022*)
*OAMB-GAL4*, *OAa2R-GAL4*, *OAb1R-GAL4*, *OAb2R-GAL4*, *OAb3R-GAL4* (*McKinney et al., 2020*)
*FRT-F3 OAMB-10XV5* (*Bakshinska et al., 2025*)
*B3RT-Tdc2-LexA*, *B3RT-vGlut-LexA*, *RSRT-STOP-RSRT-6XV5-vMAT*, *UAS-B3* and *20XUAS-R* (*Sherer et al., 2020*)
*20XUAS-B2* (*Williams et al., 2019*)
*13LexAop-6XmCherry* (*Shearin et al., 2014*)

### Stocks original to this report

*Tdc2-GAL4-DBD* (JK65C)
*B3RT-Tdc2-LexA* germline excision
*B3RT-vGlut-LexA* germline excision
*RSRT-6XV5-vMAT* germline excision
FRT-F3-OAa2R-20XV5
*FRT-F3-OAa2R-20XV5* germline inversion
B2RT-STOP-B2RT-OAb2R-40XV5
*B2RT-OAb2R-40XV5* germline excision

5-HT7-20XV5
*MHC-GCAMP8m (VK00027)*

### Entry clones

The *R4-GAL4-DBD-R3* entry clone contains a 640 bp insert encoding the GAL4 DNA binding domain. The *L1-MHC-5'Reg-R5* entry clone contains 5254 bp of upstream regulatory sequence from the *Drosophila* Myosin Heavy Chain (MHC) gene. The *L5-GCAMP8m-L2* entry clone contains a 1276 bp insert encoding GCAMP8m (*Zhang et al., 2023*). Entry clones were generated as previously described (*Petersen and Stowers, 2011*).

### Expression clones

*Tdc2-GAL4-DBD* was constructed by combining entry clones *L1-Tdc2-5'Reg-L4* and *L3-Tdc2-3'Reg-L2* (*Shearin et al., 2013*) with the *R4-GAL4-DBD-R3* entry clone and the destination vector *pDESTP10* (*Shearin et al., 2013*). *MHC-GCAMP8m* was assembled by combining entry clones *L1-MHC-5'Reg-R5* and *L5-GCAMP8m-L2* with *pDESTP10-SPEC. pDESTP10-Spec* is a derivative of *pDESTP10* in which AmpR was replaced with SpecR.

### Donor plasmid assembly

Donor plasmids were assembled starting with gene synthesis of the homology arms (Synbio). The gene synthesis also included recombinase target sites and mutations in the PAM sites associated with the guide RNAs. Homology arms typically extended ~750 bp beyond the cut sites of the guide RNAs. STOP cassettes were added via Gibson cloning. Multimerized epitope tags (*Stowers, 2025*) were added via *Asc I/Not I* restriction enzyme cloning. The *OAa2R-FRT-F3-20XV5* donor plasmid contained a pair of *FRT-F3* sites appropriately arranged for conditional expression of OAα2R-20XV5 after inversion/excision by FLP recombinase (*Fisher et al., 2017*) with the 20XV5 fused to the carboxy-terminus. The *OAb2R-B2RT-STOP-B2RT-40XV5* donor plasmid contained B2 recombinase target sites flanking a STOP cassette such that OAβ2R-40XV5 was expressed after excision of the STOP cassette with 40XV5 fused to the carboxy-terminus. The 5-HT7-20XV5 donor plasmid contained a 20XV5 multimerized epitope tag fused to the carboxy-terminus but was not designed for conditional expression.

### Guide RNA plasmid assembly

Guide RNAs were assembled in the *pCFD4* double-guide RNA plasmid as previously described (*Port et al., 2014*). Two double guide RNA plasmids were used with each donor construct for genome editing. The guide RNA plasmids and the guide RNAs they contain are as follows: *pCFD4-OAa2RGd12*-TCTA AATCAATATCAGTG and AACATAAACATCCTAAAA; *pCFD4-OAa2R-Gd34*-ACCTGTAACAGCACAC AT and GGCATGTATCAGCACTAC; *pCFD4-OAb2R-Gd56*-ATAAATGAGATTATACTC and CATGCATC TGGAATTTAT; *pCFD4-OAb2R-Gd78*-CGGCGGGCGGATCCCGCC and AGGATGGCAGGGAGTCGA; *pCFD4-5-HT7-Gd56*-TGGGCGATGAGAGGCACG and TGGGCGATGAGAGGCACG; *pCFD4-5-HT7-Gd78*-CCACGAGGACCTCCATTC and GGGTCTTCCCTAGGTTTC.

### Genome editing

For each genome edit, the donor plasmid was injected together with both *pCFD4* guide RNA plasmids into the *nos-Cas9 attP40* fly strain by Rainbow Transgenic Flies. Adult flies that were injected as embryos were crossed to third chromosome balancer stocks. Candidate progeny males from this first cross containing the *TM6b* balancer chromosome were crossed to yw; n-syb-GAL4, UAS-FLP/ TM6c (*OAa2R-FRT-F3-20XV5, yw; 20XUAS-B2; n-syb-GAL4 (OAb2R-B2RT-STOP-B2RT-40XV5)* or *yw; Ly/TM6c (5-HT7-20XV5)*) fly strain and larval progeny were screened by immunostaining. Males whose progeny exhibited positive immunostaining were subsequently crossed to a third chromosome balancer stock to establish stable lines.

### Germline excision/inversion

Germline inversion of *OAa2R-FRT-F3-20XV5* was accomplished by crossing to *yw; nos-GAL4; UAS-FLP*. Males containing all three genetic components were crossed to a third chromosome balancer stock. Individual progeny males from this cross were crossed a second time to the third chromosome

balancer stock and non-Tb larva were screened by anti-V5 immunostaining to identify germline excisions. A similar strategy was used with *OAb2R-B2RT-STOP-B2RT-40XV5* for a germline excision, except the initial cross was to *yw; nos-GAL4; UAS-B2. vGlut-LexA* was generated by crossing *B3RT-vGlut-LexA* to *yw: nos-GAL4; UAS-B3*. Progeny from this cross containing all three genetic components were crossed to a second chromosome balancer stock. Single progeny males were then crossed to a *13XLexAop-6XmCherry* reporter and larva were screened for expression in the ventral nerve cord to identify germline excision positives. A similar strategy was used to create *B3RT-Tdc2-LexA* starting with *B3RT-Tdc2-LexA* (*McKinney et al., 2020*). *RSRT-6XV5-vMAT* was generated by crossing *RSRT-STOP-RSRT-6XV5-vMAT* with *yw; nos-GAL4; UAS-R*. Progeny from this cross containing all three genetic components were crossed to a second chromosome balancer stock. Single progeny males were crossed to the second balancer a second time and larval progeny were subjected to anti-V5 immunostaining to identify germline excisions.

## Immunostaining

Male reproductive systems were dissected in PBS and fixed in 4% paraformaldehyde for 30 min with rotation. After 2X brief washes with PBS, they were incubated in PBS containing 2% Triton-X 100 and 2% BSA (PBSTB) for at least 2 hr with rotation. Subsequently, they were incubated in primary antibodies PBSTB for 1 hr with rotation, followed by overnight incubation at 4°C without rotation to prevent tissue damage. After 3X 5 min washes in PBSTB, male reproductive systems were incubated for 2–3 hr in PBSTB with secondary antibodies with rotation. After three 5 min washes in PBS, they were incubated in PBS containing Phalloidin 405 for 1 hr. After two 5 min washes in PBSTB and one 5 min wash in PBS, male reproductive systems were mounted on glass slides in Vectashield Plus (Vector Laboratories, Cat. #H-1900) using a coverslip that was glued on the edges with clear nail polish.

Primary antibodies and dilution factors: The SYN (3C11) mAb 1:20 (*Klagges et al., 1996*). Brp nc82 mAb 1:40 (deposited at the DSHB by Eric Buchner), Dlg 4F3 mAb 1:40 (*Parnas et al., 2001*), were obtained from the Developmental Studies Hybridoma Bank, created by the NICHD of the NIH and maintained at The University of Iowa, Department of Biology, Iowa City, IA 52242. Mouse anti-V5 SV5-P-K (Novus NBP2-52703) 1:200; rat anti-V5 SV5-P-K (Novus NBP2-81037) 1:200, rabbit anti-V5 SV5-P-K (Novus NBP2-52653) 1:200; Human anti-V5 (Novus NBP2-81035) 1:200; rat anti-MYC 9E10 (Novus NBP2-81020) 1:200, rabbit anti-MYC 9E10 (Novus NBP2-52636), mouse anti-MYC 4A6; (Millipore Sigma 05–724) 1:200, mouse anti-vGlut 1:10 (*Banerjee et al., 2021*), mouse anti-GFP 3E6 (Thermo Fisher A11120) 1:200; Rabbit anti-GFP (Thermo-Fisher G10362); rat anti-mCherry 16D7 (Thermo Fisher-M11217) 1:200, rabbit anti-Tdc2 pab0822-P (Covalab) 1:500; rat anti-5-HT YC5 (Abcam ab6336) 1:300.

Secondary antibodies and dilution factors: Goat anti-Mouse Alexa 405 (Thermo Fisher A31553) 1:400; Goat anti-Mouse Alexa 488 (Jackson Immunoresearch 115-545-166) 1:400; Donkey anti-Rat Alexa 488 (Jackson Immunoresearch 712-546-153) 1:400, Goat anti-Rabbit Alexa 488 (Thermo Fisher A32731) 1:400, Goat anti-Mouse Alexa 568 (Thermo Fisher A11031) 1:400, Goat anti-Rabbit Alexa 568 (Thermo Fisher A11036) 1:400, Donkey anti-Rat Alexa 568 (Thermo Fisher A78946) 1:400; Goat anti-Mouse Alexa 647 (Thermo Fisher A32728) 1:400; Goat anti-Rabbit Alexa 647 (Thermo Fisher A32733) 1:400; Goat anti-Rat Alexa 647 (Thermo Fisher A48265) 1:400; Goat anti-Mouse ATTO 647N (Rockland 610-156-121 1:100); Goat anti-Rabbit ATTO 647N (Rockland 611-156-122 1:100); Goat anti-Rat ATTO 647N (Rockland 612-156-120 1:100); Donkey anti-Human Alexa 790 (Jackson Immunoresearch 709-655-149) 1:400. Muscles were stained with Phalloidin iFluor405 (Abcam AB176752) 1:800.

Confocal microscopy was performed using a Leica Stellaris DMI8 inverted confocal scanning light microscope in the microscopy facility at the Montana State University Center for Biofilm Engineering.

## Expansion microscopy

The Ten-fold Robust Expansion Microscopy (TREx) protocol was used to expand the male ED (ED) (*Damstra et al., 2023*). Briefly, *Drosophila* male reproductive systems were dissected and fixed for at least 30 min in 4% paraformaldehyde. The EDs were then dissected out and immunostained as described above except using ATTO 647N secondary antibodies. Tissues were then incubated in Acryloyl-X solution, 1:100 dilution (Invitrogen, catalog #2604348), overnight, at room temperature and protected from light. The next day, the EDs were rinsed in PBS and incubated in TRex solution at 4° C, for 45–60 min. 0.15% ammonium persulfate and 0.15% TEMED was added and the TREx mixture

was quickly transferred to chamber slides. TREx mixture with EDs were placed in wells (1 ED per well), cover slipped and allowed to polymerize at 37 C for at least 45 min. After gel polymerization, each gel-ED was trimmed to eliminate excess gel. Gel-EDs were incubated in proteinase K (New England Biolabs, catalog #P8107S, 800 U/ml) 1:100 solution, overnight at room temperature. Expansion of gel-EDs was done the following day. Three washes in water were performed, at least 20 min per wash. DAPI (Thermofisher Scientific, catalog# D1306), 2 ug/ml, was added in the first wash to aid in tissue visualization. After the third wash, hydrogel-EDs were trimmed again and placed in chamber slides. 1 hydrogel-ED per well and cover slipped. Imaged expanded EDs in upright confocal (Leica Stellaris 8 DIVE) equipped with a 40x NA = 0.8 water objective. z Stack images were deconvoluted using the Lightning confocal function and 3D images were generated for each image. EDs typically expanded approximately tenfold.

## Colocalization analysis

Digitally deconvolved 3D images of expanded EDs were analyzed using ImageJ software (*Schindelin et al., 2012*). To exclude background in an unbiased manner, we thresholded all images by using the Otsu threshold and calculated Manders' coefficients by using the BIOP JACop.plugin. A scatter plot of green and red pixel intensities (associated with each of the two markers, Vmat and Vglut) was generated by using the colocalization function. Manders' coefficient ranges from 0 to 1, indicating no or complete colocalization. Prior to doing the colocalization analysis of each image, a region of interest (ROI) was traced using the stain from the CD8 marker. This ROI corresponded to an axon terminal.

## Mating assays

Mating assays involved placing single 3–7-day-old virgin males with single 3–7- day-old virgin females in the same well of a 12-well tissue culture plate without anesthesia. The wells of the 12-well plates were half-filled with agarose. Flies were videotaped for 3 hr. Mating assays were conducted at 25°C and 50% humidity in a room dedicated to behavioral assays. Mating was scored by viewing the videos and scorers were blind to the genotypes.

## Fertility assays

The fertility assays involved placing single 5-day-old or 30-day-old virgin males with four 3–7-day-old virgin females in fresh vials without anesthesia for 6 hr. After 6 hr, the males were removed and the four females returned to the vials for 72 hr at which point the females were removed. Six days after removal of the females, the pupae on the walls of the vials were counted. The counting was done blind to the genotypes. Fly culturing for all stages of the fertility assays were conducted at 25°C and 50% in an incubator.

## Animal use

All experiments involving *Drosophila melanogaster* were conducted in accordance with approved protocols and standard laboratory practices. *Drosophila melanogaster* is not considered a vertebrate animal and accordingly does not require Montana State University Institutional Animal Care and Use Committee (IACUC) approval.

## Complete genotypes

*Figure 1*. (B) *yw*.
*Figure 2*. (A) *yw*: *Tdc2-GAL4/+*: *UAS-CD8-mCherry/+*; (B) *yw*: *TRH-GAL4/+*: *UAS-CD8-mCherry/+*; (C) *yw*: *vGlut-GAL4/+*: *UAS-CD8-mCherry/+*; (D) *yw*: *vGlut-GAL4-DBD/+*; *Tdc2-AD/UAS-CD8-mCherry/+*; (E) *yw*; *vGlut-AD/+*; *Tdc2-GAL4-DBD, UAS-CD8-mCherry*; (F) *yw*; *TRH-AD/vGlut-GAL4-DBD*; *UAS-CD8-mCherry/+*; (G) *yw*; *vGlut-AD/+*; *TRH-GAL4-DBD/UAS-CD8-mCherry*; (H) *yw*; *Tdc2-AD/TRH-GAL4-DBD, UAS-CD8-mCherry*.
*Figure 3*. *yw*; *B3RT-Tdc2-LexA/20XUAS-6XGFP*; *TRH-GAL4/13XLexAop-6XmCherry*.
*Figure 3—figure supplement 1*.*yw*; *B3RT-vGlut-LexA/20XUAS-6XGFP*; TRH-GAL4/13XLexAop-6XmCherry.
*Figure 3—figure supplement 2*. *yw*; *B3RT-vGlut-LexA/Tdc2-GAL4*; TRH-GAL4/*20XUAS-6XGFP/13XLexAop-6XmCherry*.

*Figure 3—figure supplement 3*. (A-C) *yw; B3RT-vGlut-LexA/20XUAS-6XGFP; fru-GAL4/13XLexAop-6XmCherry.* (D) *yw; Tdc2-AD/dsx-GAL4-DBD, UAS-CD8-mCherry.* (E) *yw; TRH-AD/+; dsx-GAL4-DBD, UAS-CD8-mCherry/+.*

*Figure 4*. (A-F) *w, TBH-GFP; vGlut-40XV5/vGlut-40XV5.* (J-L) *yw; vGlut-40XV5/vGlut-40XV5.*

*Figure 4—figure supplement 1*. (A-L) *w, TBH-GFP; vGlut-40XV5/vGlut-40XV5.* (M-X) *yw; vGlut-40XV5/vGlut-40XV5.*

*Figure 4—figure supplement 2*. (A-L) *w, TBH-GFP; vGlut-40XV5/vGlut-40XV5.* (M-X) *yw; vGlut-40XV5/vGlut-40XV5.*

*Figure 4—figure supplement 3*. (A-L) *w, TBH-GFP; vGlut-40XV5/vGlut-40XV5.* (M-X) *yw; vGlut-40XV5/vGlut-40XV5.*

*Figure 4—figure supplement 4*. (A-F) *w, TBH-GFP; vGlut-40XV5/vGlut-40XV5.*

*Figure 5*. (A-L) *yw; B2RT-STOP-B2RT-40XMYC, RSRT-STOP-RSRT-6XV5-vMAT/+; TRH-GAL4, UAS-CD8-mCherry, UAS-B2, UAS-R.* (M-O) *yw; B2RT-STOP-B2RT-40XMYC, RSRT-STOP-RSRT-6XV5-vMAT/Tdc2-GAL4; UAS-CD8-mCherry, UAS-B2, UAS-R/+.*

*Figure 5—figure supplement 1*. (A-L) *yw; B2RT-STOP-B2RT-40XMYC, RSRT-STOP-RSRT-6XV5-vMAT/+; TRH-GAL4, UAS-CD8-mCherry, UAS-B2, UAS-R.* (M-O) *yw; B2RT-STOP-B2RT-40XMYC, RSRT-STOP-RSRT-6XV5-vMAT/Tdc2-GAL4; UAS-CD8-mCherry, UAS-B2, UAS-R/+.*

*Figure 5—figure supplement 2*. (A-L) *yw; B2RT-STOP-B2RT-40XMYC, RSRT-STOP-RSRT-6XV5-vMAT/+; TRH-GAL4, UAS-CD8-mCherry, UAS-B2, UAS-R.* (M-O) *yw; B2RT-STOP-B2RT-40XMYC, RSRT-STOP-RSRT-6XV5-vMAT/Tdc2-GAL4; UAS-CD8-mCherry, UAS-B2, UAS-R/+.*

*Figure 5—figure supplement 3*. (A-I) *yw; B2RT-STOP-B2RT-40XMYC, RSRT-STOP-RSRT-6XV5-vMAT/+; TRH-GAL4, UAS-CD8-mCherry, UAS-B2, UAS-R.*

*Figure 5—figure supplement 4*. (A-I) *yw; B2RT-STOP-B2RT-40XMYC, RSRT-STOP-RSRT-6XV5-vMAT/Tdc2-GAL4; UAS-CD8-mCherry, UAS-B2, UAS-R/+.*

*Figure 5—figure supplement 5*. (A-B) *yw.*

*Figure 6*. (A-I) *yw; vGlut-40XMYC, 6XV5-vMAT/vGlut-40XMYC, 6XV5-vMAT.* (A'-I') *yw; vGlut-40XMYC, 6XV5-vMAT/vGlut-40XMYC, 6XV5-vMAT.* (A"-I") *yw; vGlut-40XMYC, 6XV5-vMAT/vGlut-40XMYC, 6XV5-vMAT.*

*Figure 6—figure supplement 1*. (A-X) *yw; vGlut-40XMYC, 6XV5-vMAT/vGlut-40XMYC, 6XV5-vMAT.*

*Figure 6—figure supplement 2*. (A-V) *yw; vGlut-40XMYC, 6XV5-vMAT/vGlut-40XMYC, 6XV5-vMAT.*

*Figure 6—figure supplement 3*. (A-O) *yw; vGlut-40XMYC, 6XV5-vMAT/IA2-GFP.*

*Figure 6—figure supplement 4*. (A-O) *yw; vGlut-40XMYC, 6XV5-vMAT/IA2-GFP.*

*Figure 6—figure supplement 5*. (A, E, I) *yw; vGlut-40X5/vGlut-40XV5.* (B, F, J) *yw; 6XV5-vMAT/6XV5-vMAT.* (C, G, K) *yw; 7XMYC-vAChT/7XMYC-vAChT.* (D, H, L) *yw; 9XV5-vGAT/9XV5-vGAT.*

*Figure 7*. (A) *yw; GluRIIB-GAL4/+; UAS-CD8-mCherry.* (B) *yw; GluRIIC-GAL4/+; UAS-CD8-mCherry.*

*Figure 7—figure supplement 1*. (A) yw; OAMB-GAL4/UAS-CD8-mCherry. (B) yw; OAα2R-GAL4/UAS-CD8-mCherry. (C) yw; Oct-TyrR-GAL4/UAS-CD8-mCherry. (D) yw; OAβ1R-GAL4/UAS-CD8-mCherry. (E) yw; OAβ2R-GAL4/UAS-CD8-mCherry. (F) yw; OAβ3R-GAL4/UAS-CD8-mCherry.

*Figure 7—figure supplement 2*. (A) *yw; 5-HT71A-GAL4/+; UAS-CD8-mCherry/+.* (B) yw; *5-HT71B-GAL4/+; UAS-CD8-mCherry/+.* (C) *yw; 5-HT2A-GAL4/UAS-CD8-mCherry.* (D) *yw; 5-HT2B-GAL4/UAS-CD8-mCherry.* (E) *yw; 5-HT7-GAL4/UAS-CD8-mCherry.*

*Figure 8*. (A-R) *yw; vGlut-40XV5/GluRIIA-GFP.*

*Figure 8—figure supplement 1*. (A-R) *yw; OAMB-10XV5/OAMB-10XV5.*

*Figure 8—figure supplement 2*. (A-R) *yw; OAMB-10XV5/OAMB-10XV5.*

*Figure 8—figure supplement 3*. (A-R) *yw; OAα2R-20XV5/OAα2R-20XV5.*

*Figure 8—figure supplement 4*. (A-U) *yw; OAα2R-20XV5/OAα2R-20XV5.*

*Figure 8—figure supplement 5*. (A-R) *yw; OAβ2R-40XV5/OAβ2R-40XV5.*

*Figure 8—figure supplement 6*. (A-O) *yw; OAβ2R-40XV5/OAβ2R-40XV5.*

*Figure 8—figure supplement 7*. (A-R) *yw; OAβ2R-40XV5/OAβ2R-40XV5.*

*Figure 8—figure supplement 8*. (A-O) *yw; 5-HT7-20XV5/5-HT7-20XV5.*

*Figure 8—figure supplement 9*. (A-R) *yw; 5-HT7-20XV5/5-HT7-20XV5.*

*Figure 8—figure supplement 10*. (A-O) *yw*; *5-HT7-20XV5/5-HT7-20XV5*.

*Figure 9*. (A, E, I) male *yw*; *UAS-H2A-GFP/TRH-AD*; *VT019028-GAL4-DBD/+*. (B, F, J) female *yw*; *UAS-H2A-GFP/TRH-AD*; *VT019028-GAL4-DBD/+*. (C, G, K) male *yw*; *UAS-H2A-GFP/+*; *VT019028-GAL4-DBD/AbdB-AD*. (D, H, L) female *yw*; *UAS-H2A-GFP/+*; *VT019028-GAL4-DBD/AbdB-AD*.

*Figure 9—figure supplement 1*. (A-P) male *yw*; *UAS-H2A-GFP/TRH-AD*; *VT019028-GAL4-DBD/+*.

*Figure 9—figure supplement 2*. (A-O) male *yw*; *UAS-H2A-GFP/+*; *VT019028-GAL4-DBD/AbdB-AD*.

*Figure 9—figure supplement 3*. (A) male *yw*; *UAS-H2A-GFP/TRH-AD*; *VT019028-GAL4-DBD/+*. (B) male *yw*; *UAS-H2A-GFP/+*; *VT019028-GAL4-DBD/AbdB-AD*.

*Figure 10*. (A) column 1-*yw*; *TRH-GAL4-DBD/AbdB-AD, UAS-BONT-C/+*; column 2-*yw*; *Tdc2-GAL4-DBD/ AbdB-AD, UAS-BONT-C*. column 3-*yw*; *AbdB-AD, UAS-BONT-C/VT019028-GAL4-DBD*. column 4-*yw*; *TRH-AD/+*; *VT019028-GAL4-DBD, UAS-BONT-C*. column 5-*yw*; *AbdB-AD, UAS-BONT-C*. column 6-*UAS-BONT-C*. column 7-wildtype *Canton-S*. (B) column 1-*yw*; *AbdB-AD, UAS-BONT-C/VT019028-GAL4-DBD*. column 2-*yw*; *AbdB-AD, UAS-BONT-C*. column 3-*UAS-BONT-C*. column 1-wildtype *Canton-S*. (C) column 2-*yw*; *TRH-AD/+*; *VT019028-GAL4-DBD, UAS-BONT-C*. column 3-*yw*; *AbdB-AD, UAS-BONT-C*. column 4-*UAS-BONT-C*. column 7-wildtype *Canton-S*. (D) column 1-*yw*; *TRH-GAL4-DBD/AbdB-AD, UAS-BONT-C/+*; column 2-*yw*; *AbdB-AD, UAS-BONT-C*. column 3-*UAS-BONT-C*. column 4-wildtype *Canton-S*. (E) column 1-*yw*; *Tdc2-GAL4-DBD/ AbdB-AD, UAS-BONT-C*. column 2-*yw*; *AbdB-AD, UAS-BONT-C*. column 3-*UAS-BONT-C*. column 4-wildtype *Canton-S*.

*Figure 10—figure supplement 1*. (A) *yw*; *ProtB-GFP/+*; (B) *yw*; *ProtB-GFP/+*; *VT019028-GAL4-DBD, UAS-BONT-C*. (C and D) Canton-S wildtype.

*Figure 10—figure supplement 2*. (A) column 1-*yw*; *B3RT-vGlut-LexA/TRH-AD*, vGlut^SS1^; *VT019028-GAL4-DBD/UAS-B3*. column 2-*yw*; *B3RT-vGlut-LexA/vGlut^SS1^*; *TRH-GAL4-DBD, Tdc2-GAL4-DBD/AbdB-AD, UAS-B3*. column 3-*yw*; *B3RT-vGlut-LexA/TRH-AD*; *VT019028-GAL4-DBD/UAS-B3*. column 4-*yw*; *B3RT-vGlut-LexA/TRH-AD, vGlut^SS1^*; *UAS-B3/+*. column 5-*yw*; *B3RT-vGlut-LexA/vGlut^SS1^*; *VT019028-GAL4-DBD/UAS-B3*. (B) column 1-*yw*; *B3RT-vGlut-LexA/TRH-AD, vGlut^SS1^*; *VT019028-GAL4-DBD/UAS-B3*. column 2- *yw*; *B3RT-vGlut-LexA/TRH-AD*; *VT019028-GAL4-DBD/UAS-B3*. column 3- *yw*; *B3RT-vGlut-LexA/TRH-AD, vGlut^SS1^*; *UAS-B3/+*. column 4-*yw*; *B3RT-vGlut-LexA/vGlut^SS1^*; *VT019028-GAL4-DBD/UAS-B3*.

*Figure 10—figure supplement 3*. (A-F) *yw*; *B3RT-vGlut-LexA/TRH-AD*, vGlut^SS1^; *VT019028-GAL4-DBD/UAS-B3, LexAop-6XmCherry*.

*Video 1*. *yw*; *ProtB-GFP/+*.

*Video 2*. *yw*; *ProtB-GFP/+*; *VT019028-GAL4-DBD, UAS-BONT-C*.

*Videos 3–9*. *yw*: *MHC-GCAMP8m*.

# Acknowledgements

Stocks obtained from the Bloomington *Drosophila* Stock Center (NIH P40OD018537) were used in this study. Antibodies used in this study were obtained from the Developmental Studies Hybridoma Bank created by the NICHD of the NIH and maintained at The University of Iowa, Department of Biology. This work used a Leica Stellaris DMI8 inverted confocal scanning light microscope in the Center for Biofilm Engineering-Bioimaging and Analytical Cores, RRID:SCR_026519, which is supported by Montana State University. This work was supported by the National Institutes of Health (NIH) grant 2R01GM115510 to SJC and RSS.

# Additional information

## Funding

| Funder | Grant reference number | Author |
| --- | --- | --- |
| National Institutes of Health | 2R01GM115510 | Sarah J Certel<br>R Steven Stowers |

| Funder | Grant reference number | Author |
|--------|------------------------|--------|

The funders had no role in study design, data collection and interpretation, or the decision to submit the work for publication.

## Author contributions

Marta Chaverra, Conceptualization, Data curation, Formal analysis, Supervision, Validation, Investigation, Methodology, Writing – review and editing; John Paul Toney, Conceptualization, Data curation, Formal analysis, Validation, Investigation, Methodology, Writing – review and editing; Lizetta D Dardenne-Ankringa, Jace Tolleson Knee, Data curation, Formal analysis, Validation, Investigation, Methodology, Writing – review and editing; Ann R Morris, Joseph B Wadhams, Formal analysis, Validation, Investigation, Methodology, Writing – review and editing; Sarah J Certel, Conceptualization, Data curation, Formal analysis, Funding acquisition, Investigation, Methodology, Project administration, Writing – review and editing; R Steven Stowers, Conceptualization, Data curation, Formal analysis, Supervision, Funding acquisition, Validation, Investigation, Visualization, Methodology, Writing – original draft, Project administration, Writing – review and editing

## Author ORCIDs

R Steven Stowers [ID] https://orcid.org/0000-0002-1473-2478

Reviewer #1 (Public review): https://doi.org/10.7554/eLife.108225.3.sa1
Reviewer #2 (Public review): https://doi.org/10.7554/eLife.108225.3.sa2
Reviewer #3 (Public review): https://doi.org/10.7554/eLife.108225.3.sa3
Author response https://doi.org/10.7554/eLife.108225.3.sa4

## Additional files

### Supplementary files

MDAR checklist

Supplementary file 1. Donor Plasmid Sequences.

### Data availability

*Figure 10—source data 1* was used to generate *Figure 10*. *Figure 10—figure supplement 2—source data 1 and 2* were used to generate *Figure 10—figure supplement 2*. All plasmids and their complete DNA sequences are available as Supplementary Data. Fly strains original to this publication will be deposited at the Bloomington Drosophila Stock Center or made available upon request.

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

# Appendix 1

## Appendix 1—key resources table

| Reagent type (species) or resource | Designation | Source or reference | Identifiers | Additional information |
|---|---|---|---|---|
| Genetic reagent (*Drosophila melanogaster*) | Tdc2-GAL4 | *Cole et al., 2005* | RRID:BDSC_9313 | |
| Genetic reagent (*Drosophila melanogaster*) | GluRIIA-GAL4 | *Deng et al., 2019* | RRID:BDSC_84637 | |
| Genetic reagent (*Drosophila melanogaster*) | GluRIID-GAL4 | *Deng et al., 2019* | RRID:BDSC_84638 | |
| Genetic reagent (*Drosophila melanogaster*) | TRH-GAL4 | *Deng et al., 2019* | RRID:BDSC_84694 | |
| Genetic reagent (*Drosophila melanogaster*) | vGlut-GAL4 | *Diao et al., 2015* | RRID:BDSC_60312 | |
| Genetic reagent (*Drosophila melanogaster*) | vGlut-GAL4-DBD | *Diao et al., 2015* | | |
| Genetic reagent (*Drosophila melanogaster*) | Tdc2-AD | *Dionne et al., 2018* | : RRID:BDSC_601902 | |
| Genetic reagent (*Drosophila melanogaster*) | TRH-GAL4-DBD | *Aso et al., 2014* | RRID:BDSC_70371 | |
| Genetic reagent (*Drosophila melanogaster*) | TRH-AD | *Aso et al., 2014* | RRID:BDSC_70975 | |
| Genetic reagent (*Drosophila melanogaster*) | GluRIIB-GAL4 | *Diao et al., 2015* | RRID:BDSC_60333 | |
| Genetic reagent (*Drosophila melanogaster*) | GluRIIE-GAL4 | *Diao et al., 2015* | RRID:BDSC_60332 | |
| Genetic reagent (*Drosophila melanogaster*) | Oct-TyrR-GAL4 | *Lee et al., 2018* | RRID:BDSC_77735 | |
| Genetic reagent (*Drosophila melanogaster*) | GluRIIC-GAL4 | *Lee et al., 2018* | RRID:BDSC_83268 | |
| Genetic reagent (*Drosophila melanogaster*) | GluRIIA-GFP | *Beckers et al., 2024* | RRID:BDSC_99517 | |
| Genetic reagent (*Drosophila melanogaster*) | 5HT1A-GAL4 | *Deng et al., 2019* | RRID:BDSC_84588 | |
| Genetic reagent (*Drosophila melanogaster*) | 5HT1B-GAL4 | *Deng et al., 2019* | RRID:BDSC_84589 | |
| Genetic reagent (*Drosophila melanogaster*) | 5-HT2A-GAL4 | *Deng et al., 2019* | RRID:BDSC_84590 | |
| Genetic reagent (*Drosophila melanogaster*) | 5-HT2B-GAL4 | *Deng et al., 2019* | RRID:BDSC_84591 | |

*Appendix 1 Continued on next page*

*Appendix 1 Continued*

| Reagent type (species) or resource | Designation | Source or reference | Identifiers | Additional information |
|---|---|---|---|---|
| Genetic reagent (*Drosophila melanogaster*) | 5-HT7-GAL4 | *Deng et al., 2019* | RRID:BDSC_84592 | |
| Genetic reagent (*Drosophila melanogaster*) | TβH-GFP | *Beckers et al., 2024* | RRID:BDSC_80113 | |
| Genetic reagent (*Drosophila melanogaster*) | fru-GAL4 | Donor: Barry Dickson, Howard Hughes Medical Institute, Janelia Research Campus | RRID:BDSC_66696 | |
| Genetic reagent (*Drosophila melanogaster*) | UAS-CD8-mCherry | Donor: Frank Schnorrer, Max Planck Institute of Biochemistry | RRID:BDSC_27392 | |
| Genetic reagent (*Drosophila melanogaster*) | ProtB-GFP | *Manier et al., 2010* | RRID:BDSC_58406 | |
| Genetic reagent (*Drosophila melanogaster*) | AbdB-AD | *Diao et al., 2024* | | |
| Genetic reagent (*Drosophila melanogaster*) | vGlut-AD | *Gao et al., 2008* | | |
| Genetic reagent (*Drosophila melanogaster*) | IA2-GFP | *Yu et al., 2025* | | |
| Genetic reagent (*Drosophila melanogaster*) | UAS-BONT-C | *Han et al., 2022* | | |
| Genetic reagent (*Drosophila melanogaster*) | dsx-GAL4-DBD | *Shirangi et al., 2016* | | |
| Genetic reagent (*Drosophila melanogaster*) | vGlut-40XV5 | *Stowers, 2025* | | germline excision of STOP cassette. |
| Genetic reagent (*Drosophila melanogaster*) | vGlut-40XMYC | *Stowers, 2025* | | germline excision of STOP cassette. |
| Genetic reagent (*Drosophila melanogaster*) | B2RT-7XMYC-vAChT | *Tison et al., 2020* | | germline excision of STOP cassette. |
| Genetic reagent (*Drosophila melanogaster*) | RSRT-9XV5-vGAT | *Certel et al., 2022* | | germline excision of STOP cassette. |
| Genetic reagent (*Drosophila melanogaster*) | OAMB-GAL4 | *McKinney et al., 2020* | | |
| Genetic reagent (*Drosophila melanogaster*) | OAα2 R-GAL4 | *McKinney et al., 2020* | | |
| Genetic reagent (*Drosophila melanogaster*) | OAβ1 R-GAL4 | *McKinney et al., 2020* | | |
| Genetic reagent (*Drosophila melanogaster*) | OAβ2 R-GAL4 | *McKinney et al., 2020* | | |

*Appendix 1 Continued on next page*

*Appendix 1 Continued*

| Reagent type (species) or resource | Designation | Source or reference | Identifiers | Additional information |
|---|---|---|---|---|
| Genetic reagent (*Drosophila melanogaster*) | OAβ3 R-GAL4 | *McKinney et al., 2020* | | |
| Genetic reagent (*Drosophila melanogaster*) | FRT-F3-OAMB-10XV5 | *Bakshinska et al., 2025* | | |
| Genetic reagent (*Drosophila melanogaster*) | B3RT-Tdc2-LexA | *Sherer et al., 2020* | | |
| Genetic reagent (*Drosophila melanogaster*) | B3RT-vGlut-LexA | *Sherer et al., 2020* | | |
| Genetic reagent (*Drosophila melanogaster*) | RSRT-STOP-RSRT-6XV5-vMAT | *Sherer et al., 2020* | | |
| Genetic reagent (*Drosophila melanogaster*) | UAS-B3 | *Sherer et al., 2020* | | |
| Genetic reagent (*Drosophila melanogaster*) | 20XUAS-R | *Sherer et al., 2020* | | |
| Genetic reagent (*Drosophila melanogaster*) | 20XUAS-B2 | *Williams et al., 2019* | | |
| Genetic reagent (*Drosophila melanogaster*) | 13XLexAop-6XmCherry | *Shearin et al., 2014* | | |
| Genetic reagent (*Drosophila melanogaster*) | Tdc2-GAL4-DBD | this study | | Octopamine split-GAL4-DBD, JK65C landing site |
| Genetic reagent (*Drosophila melanogaster*) | B3RT-Tdc2-LexA germline excision | this study | | Tdc2-LexA driver |
| Genetic reagent (*Drosophila melanogaster*) | B3RT-vGlut-LexA germline excision | this study | | vGlut-LexA driver |
| Genetic reagent (*Drosophila melanogaster*) | RSRT-6XV5-vMAT germline excision | this study | | constitutively expressed 6XV5-vMAT |
| Genetic reagent (*Drosophila melanogaster*) | FRT-F3-OAα2R-20XV5 | this study | | conditionally expressible OAα2R-20XV5 |
| Genetic reagent (*Drosophila melanogaster*) | FRT-F3-OAα2R-20XV5 germline inversion | this study | | constituvely expressed OAα2R-20XV5 |
| Genetic reagent (*Drosophila melanogaster*) | B2RT-STOP-B2RT-OAβ2R-40XV5 | this study | | conditionally expressible OAα2R-40XV5 |
| Genetic reagent (*Drosophila melanogaster*) | B2RT-OAβ2R-40XV5 germline excision | this study | | constitutively expressed OAβ2R-40XV5 |
| Genetic reagent (*Drosophila melanogaster*) | 5-HT7-20XV5 | this study | | constitutively expressed 5-HT7-20XV5 |

*Appendix 1 Continued on next page*

*Appendix 1 Continued*

| Reagent type (species) or resource | Designation | Source or reference | Identifiers | Additional information |
|---|---|---|---|---|
| Genetic reagent (*Drosophila melanogaster*) | MHC-GCAMP8m | this study | | Muscle-specific expression of GCAMP8m, VK00027 landing site |
| Antibody | SYN 3 C11 supernatant | (DSHB Cat# 3 C11, RRID:AB_528479) | Mouse monoclonal | Dilution 1:20 |
| Antibody | nc82 (Brp) supernatant | (DSHB Cat# nc82, RRID:AB_231486) | Mouse monoclonal | Dilution 1:40 |
| Antibody | Dlg 4 F3 supernatant | (DSHB Cat# 4 F3 anti-discs large, RRID:AB_528203) | Mouse monoclonal | Dilution 1:40 |
| Antibody | Mouse anti-V5 | Novus NBP2-52703 | Mouse monoclonal | Dilution 1:200 |
| Antibody | Rabbit anti-V5 | Novus NBP2-52653 | Rabbit monoclonal | Dilution 1:200 |
| Antibody | Rat anti-V5 | Novus NBP2-81037 | Rat monoclonal | Dilution 1:200 |
| Antibody | Human anti-V5 | Novus NBP2-81035 | Human monoclonal | Dilution 1:200 |
| Antibody | Rat anti-MYC 9E10 | Novus NBP2-81020 | Rat monoclonal | Dilution 1:200 |
| Antibody | Rabbit anti-MYC 9E10 | Novus NBP2-52636 | Rabbit monoclonal | Dilution 1:200 |
| Antibody | Mouse anti-MYC 4 A6 | Millipore Sigma 05–724 | Mouse monoclonal | Dilution 1:200 |
| Antibody | Mouse anti-vGlut | ***Banerjee et al., 2021*** | Mouse monoclonal | Dilution 1:10 |
| Antibody | Mouse anti-GFP 3E6 | Thermo-Fisher A11120 | Mouse monoclonal | Dilution 1:200 |
| Antibody | Rabbit anti-GFP | Thermo-Fisher G10362 | Rabbit monoclonal | Dilution 1:200 |
| Antibody | Rat anti-mCherry 16D7 | ThermoFisher M11217 | Rat monoclonal | Dilution 1:200 |
| Antibody | Rabbit anti-Tdc2 | pab0822-P, Covalab | Rabbit polyclonal | Dilution 1:500 |
| Antibody | Rat anti-5-HT YC5 | Abcam ab6336 | Rat monoclonal | Dilution 1:300 |
| Antibody | Goat anti-mouse Alexa 405 | Thermo-Fisher A31553 | Goat polyclonal | Dilution 1:400 |
| Antibody | Goat anti-mouse Alexa 488 | Jackson Immunoresearch 115-545-166 | Goat polyclonal | Dilution 1:400 |
| Antibody | Donkey anti-rat Alexa 488 | Jackson Immunoresearch 712-546-153 | Donkey polyclonal | Dilution 1:400 |
| Antibody | Goat anti-rabbit Alexa 488 | Thermo-Fisher A32731 | Goat polyclonal | Dilution 1:400 |
| Antibody | Goat anti-mouse Alexa 568 | Thermo Fisher A11031 | Goat polyclonal | Dilution 1:400 |
| Antibody | Goat anti-rabbit Alexa 568 | Thermo Fisher A11036 | Goat polyclonal | Dilution 1:400 |
| Antibody | Donkey anti-rat Alexa 568 | Thermo-Fisher A78946 | Donkey polyclonal | Dilution 1:400 |
| Antibody | Goat anti-mouse Alexa 647 | Thermo-Fisher A32728 | Goat polyclonal | Dilution 1:400 |
| Antibody | Goat anti-rabbit Alexa 647 | Thermo-Fisher A32733 | Goat polyclonal | Dilution 1:400 |
| Antibody | Goat anti-rat Alexa 647 | Thermo-Fisher A48265 | Goat polyclonal | Dilution 1:400 |
| Antibody | Goat anti-mouse ATTO 647 N | Rockland 610-156-121 | Goat polyclonal | Dilution 1:100 |
| Antibody | Goat anti-rabbit ATTO 647 N | Rockland 611-156-122 | Goat polyclonal | Dilution 1:100 |
| Antibody | Goat anti-rat ATTO 647 N | Rockland 612-156-120 | Goat polyclonal | Dilution 1:100 |
| Antibody | Donkey anti-human Alexa 790 | Jackson Immunoresearch 709-655-149 | Donkey polyclonal | Dilution 1:400 |
| other | Stain, Phalloidin iFluor 405 | Abcam AB176752 | | Dilution 1:800 |

*Appendix 1 Continued on next page*

*Appendix 1 Continued*

| Reagent type (species) or resource | Designation | Source or reference | Identifiers | Additional information |
|---|---|---|---|---|
| Chemical compound, drug | Triton X-100 | Sigma-Aldrich T8787 | X-100 | |
| Chemical compound, drug | Bovine serum albumin | Lampire-cat#–7500804 | | |
| Chemical compound, drug | Acryloyl-X SE | Thermo Fisher A20770 | | |
| Chemical compound, drug | Ammonium persulfate | Sigma-Aldrich 215589 | | |
| Chemical compound, drug | Acrylamide 40% solution | Sigma-Aldrich 01697 | | |
| Chemical compound, drug | N,N'-Methylenebisacrylamide (bis) 2% solution | Sigma-Aldrich M1533 | | |
| Chemical compound, drug | DAPI | Thermo Fisher D1306 | | 2µg/ml |
| Chemical compound, drug | Proteinase K | New England Biolabs P81075 | | Dilution 1:20 |
| Software, algorithm | Image J | *Schindelin et al., 2012* | | |

