## [Editor Report · eLife Assessment]

This **important** study characterizes with rigorous methodology anatomical and functional aspects of the peripheral innervation of the Drosophila male reproductive tract. The **convincing** analysis reveals two distinct types of glutamatergic neurons that co-release either serotonin or octopamine. While serotonergic neurons are required for male fertility, octopaminergic neurons are dispensable. The work is providing invaluable insight into neurochemical control of insemination, peripheral motor control and neuromodulation in the male reproductive tract.

---

## [Referee Report · Reviewer #1 (Public review)]

Summary:

This very thorough anatomical study addresses the innervation of the Drosophila male reproductive tract. Two distinct glutamatergic neuron types were classified: serotonergic (SGNs) and octopaminergic (OGNs). By expansion microscopy, it was established that glutamate and serotonin /octopamine are co-released. The expression of different receptors for 5-HT and OA in muscles and epithelial cells of the innervation target organs was characterized. The pattern of neurotransmitter receptor expression in the target organs suggests that seminal fluid and sperm transport and emission are subjected to complex regulation. While silencing of abdominal SGNs leads to male infertility and prevents sperm from entering the ejaculatory duct, silencing of OGNs does not render males infertile.

Strengths:

The studied neurons were analysed with different transgenes and methods, as well as antibodies against neurotransmitter synthesis enzymes, building a consistent picture of their neurotransmitter identity. The careful anatomical description of innervation patterns together with receptor expression patterns if the target organs provides a solid basis for advancing the understanding how seminal fluid and sperm transport and emission are subjected to complex regulation. The functional data showing that SGNs are required for male fertility and for the release of sperm from the seminal vesicle into the ejaculatory duct is convincing.

Weaknesses:

The functional analysis of the characterized neurons is not as comprehensive as the anatomical description and phenotypic characterization was limited to simple fertility assays. It is understandable that a full functional dissection is beyond the scope of the present work. The paper contains experiments showing neuron-independent peristaltic waves in the reproductive tract muscles, which are thematically not very well integrated into the paper. Although very interesting, one wonders if these experiments would not fit better into a future work that also explores these peristaltic waves and their interrelation with neuromodulation mechanistically.

Comments on revisions:

The manuscript has improved after fixing many small issues/errors. The new sections in the discussion are likewise adding to the quality of the manuscript.

---

## [Referee Report · Reviewer #2 (Public review)]

Summary:

Cheverra et al. present a comprehensive anatomical and functional analysis of the motor neurons innervating the male reproductive tract in *Drosophila melanogaster*, addressing a gap in our understanding of the peripheral circuits underlying ejaculation and male fertility. They identify two classes of multi-transmitter motor neurons-OGNs (octopamine/glutamate) and SGNs (serotonin/glutamate)-with distinct innervation patterns across reproductive organs. The authors further characterize the differential expression of glutamate, octopamine, and serotonin receptors in both epithelial and muscular tissues of these organs. Behavioral assays reveal that SGNs are essential for male fertility, whereas OGNs and glutamatergic transmission are dispensable. This work provides a high-resolution map linking neuromodulatory identity to organ-specific motor control, offering a valuable framework to explore the neural basis of male reproductive function.

Strengths:

Through the use of an extensive set of GAL4 drivers and antibodies, this work successfully and precisely defines the neurons that innervate the male reproductive tract, identifying the specific organs they target and the nature of the neurotransmitters they release. It also characterizes the expression patterns and localization of the corresponding neurotransmitter receptors across different tissues. The authors describe two distinct groups of dual-identity neurons innervating the male reproductive tract: OGNs, which co-express octopamine and glutamate, and SGNs, which co-express serotonin and glutamate. They further demonstrate that the various organs within the male reproductive system differentially express receptors for these neurotransmitters. Based on these findings, the authors propose that a single neuron capable of co-releasing a fast-acting neurotransmitter along side a slower-acting one may more effectively synchronize and stagger events that require precise timing. This, together with the differential expression of ionotropic glutamate receptors and metabotropic aminergic receptors in postsynaptic muscle tissue, adds an additional layer of complexity to the coordinated regulation of fluid secretion, organ contractility, and directional sperm movement-all contributing to the optimization of male fertility.

Weaknesses:

One potential limitation of the study is the absence of information regarding the number of individuals examined for the various characterizations, which may weaken the strength of the conclusions. Another limitation may be the lack of quantitative analyses in the colocalization and morphological differentiation experiments. Nevertheless, the authors have indicated that such quantifications will be provided in a forthcoming publication; therefore, this should be considered only a partial limitation, as it is expected to be addressed in the near future.

Wider context:

This study delivers the first detailed anatomical map connecting multi-transmitter motor neurons with specific male reproductive structures. It highlights a previously unrecognized functional specialization between serotonergic and octopaminergic pathways and lays the groundwork for exploring fundamental neural mechanisms that regulate ejaculation and fertility in males. The principles uncovered here may help explain how males of Drosophila and other organisms adjust reproductive behaviors in response to environmental changes. Furthermore, by shedding light on how multi-transmitter systems operate in reproductive control, this model could provide insights into therapeutic targets for conditions such as male infertility and prostate cancer-where similar neuronal populations are involved in humans. Ultimately, this genetically accessible system serves as a powerful tool for uncovering how multi-transmitter neurons orchestrate coordinated physiological actions necessary for the functioning of complex organs.

---

## [Referee Report · Reviewer #3 (Public review)]

Summary:

This work provides an overview of the motor neuron landscape in the male reproductive system. Some work had been done to elucidate the circuits of ejaculation in the spine, as well as, the cord but this work fills a gap of knowledge at the level of the reproductive organs. Using complementary approaches the authors show that there are two types of motor neurons that are mutually exclusive: neurons that co-express octopamine and glutamate and neurons that co-express serotonin and glutamate. They also show evidence that both types of neurons express large dense core vesicles indicating that neuropeptides play a role in male fertility. This paper provides a thorough characterization of expression of the different glutamate, octopamine and serotonin receptors in the different organs and tissues of the male reproductive system. The differential expression in different tissues and organs allows building initial theories on the control of emission and expulsion. Additionally, the authors characterize the expression of synaptic proteins and the neuromuscular junction sites. On a mechanistic level, the authors show that neither octopamine/glutamate neuron transmission nor glutamate transmission in serotonin/glutamate neurons are required for male fertility. This final result is quite surprising and opens up many questions on how ejaculation is coordinated.

Strengths:

This work fills an important gap on characterization of innervation of the male reproductive system by providing an extensive characterization of the motor neurons and the potential receptors of motor neuron release.The authors show convincing evidence of glutamate/monoamine co-release and of mutual exclusivity of serotonin/glutamate and octopamine/glutamate neurons.

Weaknesses:

The experiment looking at peristaltic waves in the male organs is missing labeling of the different regions and quantification of the observed waves.

---

## [Author Response]

The following is the authors’ response to the original reviews.

**Reviewer #1 (Public review):**
Summary:This very thorough anatomical study addresses the innervation of the Drosophila male reproductive tract. Two distinct glutamatergic neuron types were classified: serotonergic (SGNs) and octopaminergic (OGNs). By expansion microscopy, it was established that glutamate and serotonin /octopamine are co-released. The expression of different receptors for 5-HT and OA in muscles and epithelial cells of the innervation target organs was characterized. The pattern of neurotransmitter receptor expression in the target organs suggests that seminal fluid and sperm transport and emission are subjected to complex regulation. While silencing of abdominal SGNs leads to male infertility and prevents sperm from entering the ejaculatory duct, silencing of OGNs does not render males infertile.Strengths:The studied neurons were analysed with different transgenes and methods, as well as antibodies against neurotransmitter synthesis enzymes, building a consistent picture of their neurotransmitter identity. The careful anatomical description of innervation patterns together with receptor expression patterns of the target organs provides a solid basis for advancing the understanding of how seminal fluid and sperm transport and emission are subjected to complex regulation. The functional data showing that SGNs are required for male fertility and for the release of sperm from the seminal vesicle into the ejaculatory duct is convincing.Weaknesses:The functional analysis of the characterized neurons is not as comprehensive as the anatomical description, and phenotypic characterization was limited to simple fertility assays. It is understandable that a full functional dissection is beyond the scope of the present work. The paper contains experiments showing neuron-independent peristaltic waves in the reproductive tract muscles, which are thematically not very well integrated into the paper. Although very interesting, one wonders if these experiments would not fit better into a future work that also explores these peristaltic waves and their interrelation with neuromodulation mechanistically.
**Reviewer #2 (Public review):**
Summary:Cheverra et al. present a comprehensive anatomical and functional analysis of the motor neurons innervating the male reproductive tract in *Drosophila melanogaster*, addressing a gap in our understanding of the peripheral circuits underlying ejaculation and male fertility. They identify two classes of multi-transmitter motor neurons-OGNs (octopamine/glutamate) and SGNs (serotonin/glutamate)-with distinct innervation patterns across reproductive organs. The authors further characterize the differential expression of glutamate, octopamine, and serotonin receptors in both epithelial and muscular tissues of these organs. Behavioral assays reveal that SGNs are essential for male fertility, whereas OGNs and glutamatergic transmission are dispensable. This work provides a high-resolution map linking neuromodulatory identity to organ-specific motor control, offering a valuable framework to explore the neural basis of male reproductive function.Strengths:Through the use of an extensive set of GAL4 drivers and antibodies, this work successfully and precisely defines the neurons that innervate the male reproductive tract, identifying the specific organs they target and the nature of the neurotransmitters they release. It also characterizes the expression patterns and localization of the corresponding neurotransmitter receptors across different tissues. The authors describe two distinct groups of dual-identity neurons innervating the male reproductive tract: OGNs, which co-express octopamine and glutamate, and SGNs, which co-express serotonin and glutamate. They further demonstrate that the various organs within the male reproductive system differentially express receptors for these neurotransmitters. Based on these findings, the authors propose that a single neuron capable of co-releasing a fast-acting neurotransmitter alongside a slower-acting one may more effectively synchronize and stagger events that require precise timing. This, together with the differential expression of ionotropic glutamate receptors and metabotropic aminergic receptors in postsynaptic muscle tissue, adds an additional layer of complexity to the coordinated regulation of fluid secretion, organ contractility, and directional sperm movement-all contributing to the optimization of male fertility.Weaknesses:The main weakness of the manuscript is the lack of detail in the presentation of the results. Specifically, all microscopy image figures are missing information about the number of samples (N), and in the case of colocalization experiments, quantitative analyses are not provided. Additionally, in the first behavioral section, it would be beneficial to complement the data table with figures similar to those presented later in the manuscript for consistency and clarity.Wider context:This study delivers the first detailed anatomical map connecting multi-transmitter motor neurons with specific male reproductive structures. It highlights a previously unrecognized functional specialization between serotonergic and octopaminergic pathways and lays the groundwork for exploring fundamental neural mechanisms that regulate ejaculation and fertility in males. The principles uncovered here may help explain how males of Drosophila and other organisms adjust reproductive behaviors in response to environmental changes. Furthermore, by shedding light on how multi-transmitter systems operate in reproductive control, this model could provide insights into therapeutic targets for conditions such as male infertility and prostate cancer, where similar neuronal populations are involved in humans. Ultimately, this genetically accessible system serves as a powerful tool for uncovering how multi-transmitter neurons orchestrate coordinated physiological actions necessary for the functioning of complex organs.
**Reviewer #3 (Public review):**
Summary:This work provides an overview of the motor neuron landscape in the male reproductive system. Some work had been done to elucidate the circuits of ejaculation in the spine, as well as the cord, but this work fills a gap in knowledge at the level of the reproductive organs. Using complementary approaches, the authors show that there are two types of motor neurons that are mutually exclusive: neurons that co-express octopamine and glutamate and neurons that co-express serotonin and glutamate. They also show evidence that both types of neurons express large dense core vesicles, indicating that neuropeptides play a role in male fertility. This paper provides a thorough characterization of the expression of the different glutamate, octopamine, and serotonin receptors in the different organs and tissues of the male reproductive system. The differential expression in different tissues and organs allows building initial theories on the control of emission and expulsion. Additionally, the authors characterize the expression of synaptic proteins and the neuromuscular junction sites. On a mechanistic level, the authors show that neither octopamine/glutamate neuron transmission nor glutamate transmission in serotonin/glutamate neurons is required for male fertility. This final result is quite surprising and opens up many questions on how ejaculation is coordinated.Strengths:This work fills an important gap in the characterization of innervation of the male reproductive system by providing an extensive characterization of the motor neurons and the potential receptors of motor neuron release. The authors show convincing evidence of glutamate/monoamine co-release and of mutual exclusivity of serotonin/glutamate and octopamine/glutamate neurons.Weaknesses:(1) Often, it is mentioned that the expression is higher or lower or regional without quantification or an indication of the number of samples analysed.(2) The experiment aimed at tracking sperm in the male reproductive system is difficult to interpret when it is not assessed whether ejaculation has occurred.(3) The experiment looking at peristaltic waves in the male organs is missing labeling of the different regions and quantification of the observed waves.
**Recommendations for the authors:**

**Reviewer #1 (Recommendations for the authors):**
(1) While the peripheral innervations are very carefully described, it is not clear to which SGNs and OGNs (i.e., cell bodies in the central nervous system) these innervations belong. Are SV, AG, and ED innervated by branches of one neuron or by separate neurons? Multi-color flip-out experiments could provide an answer to this.

We agree this is important and are planning these experiments for follow-up study.

(2) In contrast, for the analysis of the VT19028 split line (Figure 9), only vnc and cell body images are shown. How do the arborisations of these split combinations look in the periphery? Are the same reproductive organs innervated as shown in Figure 2?

Figure 9S3 was inadvertently omitted from the initial submission. That figure is now included and shows that the VT019028 split broadly innervates the SV, AG, and ED.

(3) In the discussion, I think it would be helpful to offer some potential explanations for the role of octopaminergic and glutamatergic signaling. If not required for basic fertility, they probably have some other role.

Thank you, we have included speculation in the Discussion section "Potential for adaptation to environment".

(4) Line 543: Figure 8S4 E, (not 8E).

Correction made.

**Reviewer #2 (Recommendations for the authors):**
(1) Line 213-217Comment:The use of "significantly less expression" may be misleading, as no quantification or statistical analysis is provided to support this comparison.Suggestion:Consider using a more neutral term, such as "markedly less" or "noticeably less," unless quantitative data and statistical analysis are included to substantiate the claim.

Good recommendation.This suggestion has been incorporated.

(2) Line 264-267Comment:The observation regarding the distinct morphology of SGNs and OGNs is interesting and could strengthen the argument regarding functional differences.Suggestion:Consider including a quantification of morphological complexity (e.g., branching) to support the claim. A method such as Sholl analysis (Sholl, 1953), as adapted in Fernández et al., 2008, could be applied.

This is a good suggestion, and we will consider it as part of a follow-up study.

(3) Line 269-271Comment:The anatomical context of the observation is not explicitly stated.Suggestion:Add "in the ED" for clarity: "With the TRH-GAL4 experiment in the ED, vGlut-40XMYC (Figure 5S1, A and E) and 6XV5-vMAT (Figure 5S1, B and F) were both present with a highly overlapping distribution (Figure 5S1, I)."

Suggestion has been incorporated.

(4) Line 275-276Comment:The claim about the reduced ability to distinguish SGNs and OGNs in the ED would benefit from quantitative support.Suggestion:Include a morphological comparison or quantification between SGNs and OGNs in the ED and SV to reinforce this point.

Certain information on morphological comparison can be inferred within the images themselves, and we will include quantitation in a follow-up study.

(5) Line 277-279Comment:As with line 269, the anatomical site could be specified more clearly.Suggestion:Rephrase as: "With the Tdc2-GAL4 experiment in the ED, vGlut-40XMYC (Figure 5S1, M and Q) and 6XV5-vMAT (Figure 5S1, N and R) were both observed in a highly overlapping distribution (Figure 5S1, U)."

Suggestion has been incorporated.

(6) Line 348-350Comment:The phrase "significantly higher density" implies a statistical comparison that is not shown.Suggestion:If no quantification is provided, replace with a qualitative term such as "visibly higher" or "notably more dense." Alternatively, add a quantitative analysis with statistical testing to justify the use of "significantly."

Suggestion has been incorporated.

(7) Lines 415-458 (Section comment)Comment:There appears to be differential localization of neurotransmitter receptor expression (glutamate in muscle vs. 5-HT in epithelium or neurons), which could have functional implications.Suggestion:Expand this section to briefly discuss the differential localization patterns of these receptors and potential implications for signal transduction in male reproductive tissues.(8) Lines 638-682 (Section comment)Comment:The table summarizing fertility phenotypes would be more informative with additional detail on experimental outcomes.Suggestion:Add a column showing the number of fertile males over the total tested (e.g., "n fertile / n total"). Also, clarify whether the fertility assays are identical to those reported in Figure 10S2, and whether similar analyses were conducted for females. Consider including a figure summarizing fertility results for all genotypes listed in the table, similar to Figure 10S2.

The fertility tests reported in Table 1 were separate from those reported in Figure 10S2. For these tests, the results were clear-cut with 100% of males and females reported as infertile exhibiting the infertile phenotype. For the males and females reported as fertile, it was also clear-cut with nearly 100% showing fertility at a high level. In subsequent figures we attempted to assess degrees of fertility.

(9) Line 724-727Comment:There seems to be a mistake in the identification of the driver lines used to silence OA neurons. Also, figure references might be incorrect.Suggestion:The OA neuron driver line should be corrected to "Tdc2-GAL4-DBD ∩ AbdB-AD" instead of TRH-GAL4. Additionally, the figure references should be verified; specifically, the letter "B" (in "Figure 10B, D" and "10B, E") appears to be unnecessary or misplaced.

Thanks for catching this, the corrections have been made.

(10) Line 872-877Comment:The discussion on the co-release of fast-acting glutamate and slower aminergic neurotransmitters is interesting and well-articulated. However, it remains somewhat disconnected from the behavioral findings.Suggestion:Consider linking this proposed mechanism to the results observed in the mating duration assays. For instance, the sequential action of neurotransmitters described here could potentially underlie the prolonged mating observed when specific neuromodulators are active, helping to functionally integrate molecular and behavioral data.(11) Line 926-928Comment:The interpretation of 5-HT7 receptor expression in the sphincter is compelling, suggesting a role in regulating its function. However, this anatomical observation could be further contextualized with the functional data.Suggestion:It may strengthen the interpretation to explicitly connect this finding with the fertility assays, where SGNs - presumably acting via serotonergic signaling - are shown to be necessary for male fertility. This would support a functional role for 5-HT7 in reproductive success via sphincter regulation.

This has been added.

(12) Figure 1Comment:The figure legend is generally clear, but could benefit from more consistency and precision in the color-coded labeling. Additionally, the naming of some structures could be more explicit.Suggestion:Revise the figure and the legend as follows:Figure 1. The Drosophila male reproductive system. (A) Schematic diagram showing paired testes (colour), SVs (green), AGs (purple), Sph (red), ED (gray), and EB (colour). (B) Actual male reproductive system. Te - testes, SV - seminal vesicle, AG - accessory gland, Sph - singular sphincter, ED - ejaculatory duct, EB - ejaculatory bulb. Scale bar: 200 µm.

This suggestion has been incorporated.

(13) Figure 3S2Comment:There appears to be a typographical error in the description of the genotypes, which may lead to confusion.Suggestion:Correct the legend to reflect the appropriate genotypes:Figure 3S2. Expression of vGlut-LexA and Tdc2-GAL4 in the Drosophila male reproductive system. (A, D, G, J, M, P) vGlut-LexA, LexAop-6XmCherry; (B, E, H, K, N, Q) Tdc2-GAL4, UAS-6XGFP; (C, F, I, L, O, R) Overlay. Scale bars: O - 50 µm; R - 10 µm.

The corrections have been made.

(14) Figure 3S3Comment:The genotypes for panels D and E appear to be incomplete; the DBD component of the split-GAL4 drivers is missing.Suggestion:Update the figure legend to:Figure 3S3. Fruitless and Doublesex expression in the Drosophila male reproductive system. (A) fru-GAL4, UAS-6XGFP; (B) vGlut-LexA, LexAop-6XmCherry; (C) Overlay; (D) Tdc2-AD ∩ dsx-GAL4-DBD; (E) TRH-AD ∩ dsx-GAL4-DBD. Scale bar: 200 µm.

The corrections have been made.

(15) Figure 4S4Comment:There is a repeated segment in the figure legend, which makes it unclear and redundant.Suggestion:Edit the legend to remove the duplicated lines:Figure 4S4. Expression of vGlut, TβH-GFP, and 5-HT at the junction of the SV and AGs with the ED of the Drosophila male reproductive system. (A) vGlut-40XV5; (B) TβH-GFP; (C) 5-HT; (D) vGlut-40XV5, TβH-GFP overlay; (E) vGlut-40XV5, 5-HT overlay; (F) TβH-GFP, 5-HT overlay. Scale bar: 50 µm.

The correction has been made.

(16) Figure 6S5Comment:Within this figure, the orientation and/or scale of the tissue varies noticeably between individual panels, making it difficult to directly compare the different experimental conditions.Suggestion:For improved clarity and interpretability, consider standardizing the orientation and size of the tissue shown across all panels within the figure. Consistent presentation will facilitate direct comparisons between treatments or genotypes.

There is often variation in the size of the male reproductive organs. They were all acquired at the same magnification. The only point of this figure is there is no vGAT or vAChT at these NMJs and the result is unambiguously negative.

(17) Figure 10Comment:Panel A appears redundant, as it shows the same information as the other panels but without indicating statistical significance.Suggestion:Consider removing panel A and keeping only the remaining four graphs, which include relevant statistical comparisons and clearly show significant differences.

We realize there is some redundancy of panel A with the other panels, but we feel there is value in having all the genotypes in a single panel for comparison.

**Reviewer #3 (Recommendations for the authors):**
Here are some suggestions to improve the manuscript:(1) Prot B GFP experiment: the authors should explain better the time chosen to look at the sperm content of the male reproductive system. At 10 minutes, it is expected that the male has already ejaculated, and therefore, a failure to ejaculate would result in more sperm in the reproductive system, not less. Since we are not certain when the male ejaculates, it would be important to do the analysis at different time points.

In the Prot-GFP experiments, the 10-minute time point was chosen because we nearly always observe sperm in the ejaculatory duct of control males. In the experimental males, we never observed sperm in the ejaculatory duct at this time point. Also, no Prot-GFP sperm were observed in the reproductive tract of females mated to experimental males even when mating was allowed to go to completion, while abundant sperm were found in females mated to Prot-GFP controls. Figure 10S1 has been updated to include Images of these female reproductive systems. The results showing the absence of Prot-GFP sperm in the female reproductive tract mated to experimental males indicates sperm transfer in these males isn't occurring earlier during the copulation process than in control males and that we didn't miss it by only examining at the ejaculatory duct.

(2) Discuss what may be the role of the octopamine/glutamate neurons and glutamate transmission in serotonin/glutamate neurons in the male reproductive system, given that they are not required for fertility (at least under the context in which it was tested). It is quite a striking result that deserves some attention.

We agree it is a surprising result and have included speculation on the role of glutamate and octopamine in male reproduction in the Discussion section "Potential for adaptation to environment".

(3) Very important:(a) Figure 3 is present in the Word document but not the PDF.(b) Figure 9S3 is not present(c) In (Figure 5 X), the legend does not correspond to the panel.

All of these corrections have been made.

(4) Other suggestions:(a) A summary schematic (or several) of the findings would make it an easier read.(b) Explain why the ejaculatory bulb was left out of the analysis.(c) Explain in the main text some of the tools, such as, BONT-C and the conditional vGlut mutation.